# Neural circuitry coordinating male copulation

**Hania J Pavlou[1]\*[†], Andrew C Lin[1,2], Megan C Neville[1], Tetsuya Nojima[1], Fengqiu Diao[3], Brian E Chen[4,5], Benjamin H White[3], Stephen F Goodwin[1]\***

[1]Centre for Neural Circuits and Behaviour, University of Oxford, Oxford, United Kingdom; [2]Department of Biomedical Science, University of Sheffield, Sheffield, United Kingdom; [3]Laboratory of Molecular Biology, National Institute of Mental Health, Bethesda, United States; [4]Department of Medicine, McGill University, Montréal, Canada; [5]Department of Neurology and Neurosurgery, McGill University, Montréal, Canada

**\*For correspondence:** hania. pavlou@imm.ox.ac.uk (HJP); stephen.goodwin@dpag.ox.ac.uk (SFG)

**Present address:** [†]Weatherall Institute of Molecular Medicine, MRC Centre for Computational Biology, University of Oxford, Oxford, United Kingdom

**Competing interests:** The authors declare that no competing interests exist.

**Abstract** Copulation is the goal of the courtship process, crucial to reproductive success and evolutionary fitness. Identifying the circuitry underlying copulation is a necessary step towards understanding universal principles of circuit operation, and how circuit elements are recruited into the production of ordered action sequences. Here, we identify key sex-specific neurons that mediate copulation in *Drosophila*, and define a sexually dimorphic motor circuit in the male abdominal ganglion that mediates the action sequence of initiating and terminating copulation. This sexually dimorphic circuit composed of three neuronal classes – motor neurons, interneurons and mechanosensory neurons – controls the mechanics of copulation. By correlating the connectivity, function and activity of these neurons we have determined the logic for how this circuitry is coordinated to generate this male-specific behavior, and sets the stage for a circuit-level dissection of active sensing and modulation of copulatory behavior.

## Introduction

All animals must continuously sequence and coordinate behaviors appropriate to both their environment and internal state if they are to survive and reproduce. Dissecting the neural substrates that initiate, organize, and terminate these behavioral sequences is critical to understanding behavior. Here, we use *Drosophila* male courtship behavior as a model to address how a compact circuit coordinates these critical action sequences. *Drosophila* male courtship behavior is a multi-step goal-directed behavior that has evolved to achieve reproductive success (*Villella and Hall, 2008*). Typically, the male follows the female, taps her with his forelegs, contacts her genitalia with his mouthparts, sings a species-specific courtship song, and bends his abdomen to copulate (*Villella and Hall, 2008*). Successful execution of these discrete sequential motor programs requires continuous integration of multiple sensory cues from the female.

Copulation, the direct objective of courtship, is a highly conserved and essential behavioral step for most animals. Copulation itself also consists of an ordered behavioral progression: first, the male engages external genital structures to grasp the female; then he extrudes the intromittent organ, the aedeagus, and initiates copulation (*Kamimura, 2010*). The male maintains this posture for 15 min or more while transferring a mixture of sperm and seminal fluid to the female (*Villella and Hall, 2008*). Finally, the male terminates copulation by sequential uncoupling of his genitals and detachment from the female. Failure at any point in this complex action sequence, from courtship to termination of copulation, may prevent reproduction. It follows that the neural substrates of male courtship behavior have evolved to carry out the precise action sequences of copulation.

**eLife digest** Idioms and love songs often euphemistically refer to "the birds and the bees". Yet for neurobiologists interested in uncovering basic facts about sex and reproduction, the fruit fly has proven much more informative.

Male fruit flies court females with a series of "hard-wired" or genetically programmed behaviors. One gene called *doublesex* generates differences in the anatomy and behavior of males and females in many animal species. In male fruit flies, the *doublesex* gene is active in roughly 650 neurons, with specific groups of cells controlling distinct steps of the courtship ritual. However, it was not understood how the different steps involved in copulation were coordinated to ensure a successful mating.

Pavlou et al. have now identified a circuit of *doublesex*-expressing neurons that controls copulation itself. The circuit, which is in the fruit fly's equivalent of the spinal cord, is made up of three types of neurons: motor neurons, inhibitory interneurons and mechanosensory neurons. The motor neurons coordinate the joining of the male's genitals with those of the female. The inhibitory interneurons promote the release of the male's genitals by opposing the motor neurons, while the mechanosensory neurons possibly coordinate the activity of the other neurons to generate the correct sequence of events needed for copulation. Pavlou et al. also showed that the mechanism that controls how the male attaches to and detaches from the female is independent of ejaculation, indicating that the mechanics of copulation are separate from those of reproduction.

A future challenge will be to understand how command centres in the brain combine these signals with sensory feedback to enable males to execute and modify their copulation-related behaviors. Identifying neural circuits that drive behaviors in fruit flies provide insights into the universal principles by which a nervous system can coordinate complex motor behaviors such as walking and flying.

Male reproductive behaviors are controlled by neurons expressing the two key sex determination genes, *doublesex* (*dsx*) and *fruitless* (*fru*) (*Pavlou and Goodwin, 2013*), the male-specific isoforms of which (Fru$^M$ and Dsx$^M$) specify male-specific neurons (*Billeter et al., 2006*; *Rideout et al., 2007*, *2010*). Manipulating the activity of some, or all, of these neurons profoundly alters male courtship and copulatory behaviors (*Stockinger et al., 2005*; *Billeter et al., 2006*; *Rideout et al., 2010*; *Kohatsu et al., 2011*; *Pan et al., 2011*; *von Philipsborn et al., 2011*; *Tran et al., 2014*). For example, activating *fru*- or *dsx*-expressing neurons in males elicits all courtship and copulatory behaviors (*Pan et al., 2011*), while inhibiting all ~650 *dsx* neurons in males blocks all courtship and copulatory behaviors (*Rideout et al., 2010*). Interestingly, activation of *dsx* circuitry in males lacking *fru*-expression (*fru$^M$*-null) elicits robust courtship and copulatory behaviors (*Pan et al., 2011*), suggesting that *dsx*-specified circuitry encompasses the fundamental neural substrates underlying both courtship and copulation (*Rideout et al., 2010*; *Pan et al., 2011*). However, it remains unknown, which of these *dsx* neurons control male copulation.

Gynandromorph studies suggest that copulation is regulated by neurons in both the central brain and the abdominal ganglion (Abg), the most posterior region of the ventral nerve cord (VNC) (*Hall, 1979*; *Ferveur and Greenspan, 1998*). Indeed, optogenetic activation of *dsx*-expressing pC2l cluster of neurons in the male brain has been shown to induce attempts to copulate (*Kohatsu and Yamamoto, 2015*). However, activating *dsx* neurons in solitary headless males elicits abdominal curling (*Pan et al., 2011*) and decapitating males *in copulo* has little effect on the duration of copulation (*Tayler et al., 2012*; *Crickmore and Vosshall, 2013*), suggesting the Abg can direct many copulatory behaviors independent of any input from the brain. It is therefore likely that descending signals from the brain serve to initiate copulation by triggering a local circuit module in the Abg that in turn coordinates copulation, but the nature of this Abg circuit has remained unclear.

While the expression of *fru* and *dsx* in sensory-, inter- and motor neurons suggests that they are organized into circuit elements capable of receiving, processing and transferring information that controls sexual behavior (*Pavlou and Goodwin, 2013*), surprisingly little is known about the core circuit elements encompassing copulation. To investigate the organizational principles of the sex-

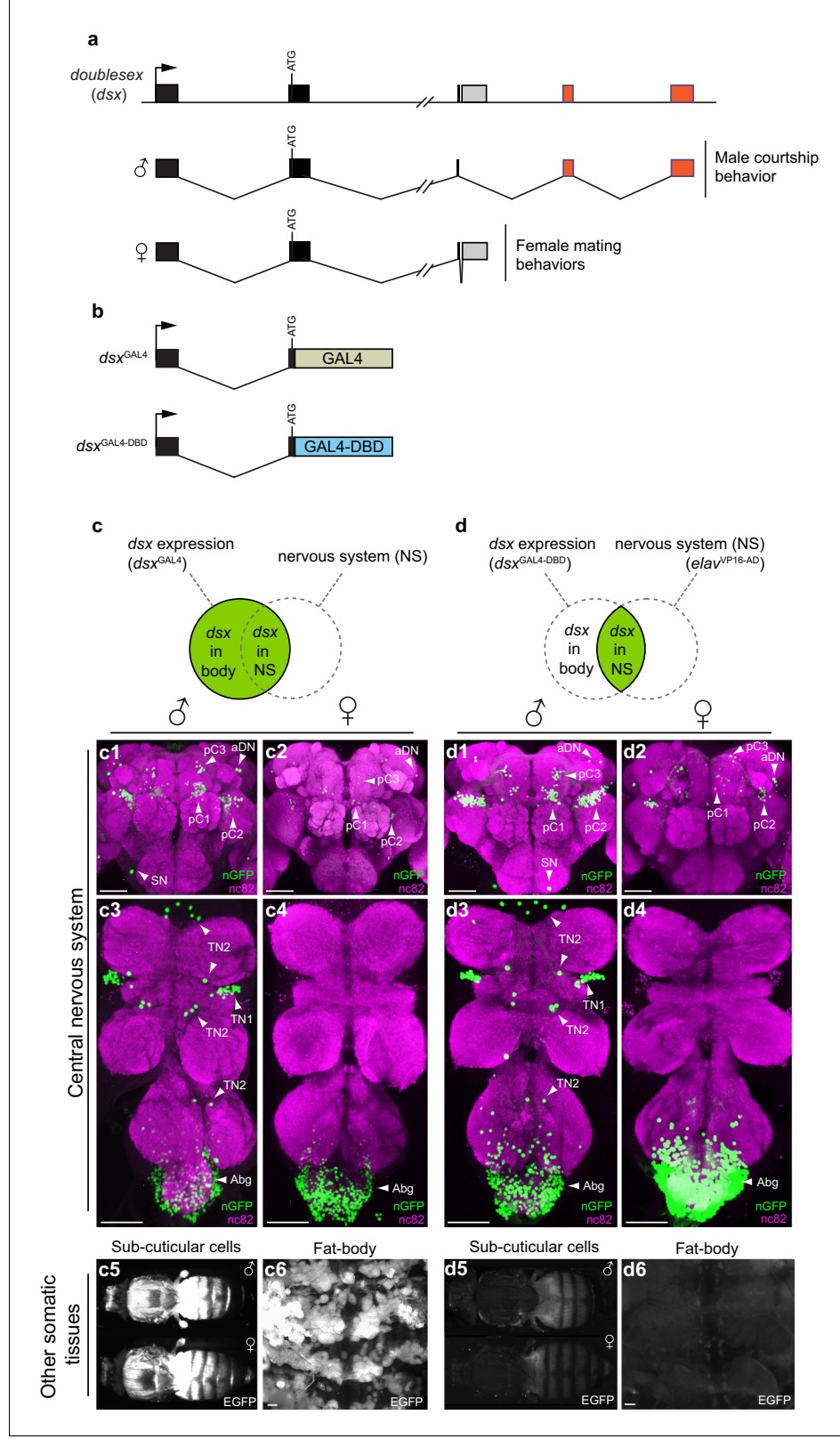

**Figure 1.** Spatial restriction of GFP expression to *dsx* neurons using novel *dsx* Split-GAL4 allele. (**a**) Schematic of *doublesex* (*dsx*) gene and male and female predicted transcripts. Arrows indicate transcriptional start sites. Colored boxes depict non-sex-specific (black) and sex-specific (red: male and grey: female) exons. (**b**) Schematic of $dsx^{GAL4}$ and $dsx^{GAL4-DBD}$ knock-in alleles. (**c**) GFP expression in five day-old males and females driven by $dsx^{GAL4}$.

*Figure 1 continued on next page*

*Figure 1 continued*

(**c1–4**) *dsx*<sup>GAL4</sup> driving *UAS-nuclear GFP* (nGFP) in (**c1**) adult male brain and (**c3**) VNC and (**c2**) adult female brain and (**c4**) VNC. (**c5–6**) Epifluorescence images of *dsx*<sup>GAL4</sup> driving *UAS-2XEGFP* (EGFP) in (**c5**) adult male and female whole-fly preparations revealing EGFP expression in sub- and peri-cuticular cells and (**c6**) adult male filleted dorsal abdominal wall revealing EGFP expression in the adult fat body. (**d**) GFP expression in five day-old males and females driven by *dsx*<sup>GAL4-DBD</sup> combined with pan-neuronal *elav*<sup>VP16-AD</sup> hemidriver. (**d1–4**) *dsx*<sup>GAL4-DBD</sup>/*elav*<sup>VP16-AD</sup> (referred to as *dsx/elav* in text) driving *UAS-nGFP* in (**d1**) adult male brain and (**d3**) VNC and (**d2**) adult female brain and (**d4**) VNC. Epifluorescence images of *dsx*<sup>GAL4DBD</sup>/*elav*<sup>VP16-AD</sup> driving *UAS-2XEGFP* in (**d5**) adult male and female whole-fly preparations revealing no EGFP expression in sub- and peri-cuticular cells and (**d6**) adult male filleted dorsal abdominal wall revealing no EGFP expression in the adult fat body. nGFP realized with anti-GFP antibody (green) and neuropil counterstained with nc82 (magenta). EGFP realized with anti-GFP antibody (white). (**c1–4**) and (**d1–d4**) views are ventral, with anterior up. Scale bar = 50 μm.

The following figure supplement is available for figure 1:

**Figure supplement 1.** *dsx*-expressing neurons specify male and female sexual behaviors.

specific circuits underlying copulation, we used a Split-GAL4 intersectional approach (*Luan et al., 2006*) to identify *dsx* neurons within the Abg that express the major excitatory and inhibitory neurotransmitters in *Drosophila*. We found that *dsx* glutamatergic motor neurons innervate muscles of the genitalia and enable genital attachment and intromission; *dsx* GABAergic inhibitory neurons mediate genital uncoupling likely by inhibition of key motor neurons; and *dsx* mechanosensory neurons of the genitalia innervate and activate both *dsx* GABAergic and *dsx* glutamatergic neurons in the Abg. These results suggest a model in which *dsx* configures a sexually dimorphic sensorimotor circuit which allows the male to successfully execute the correct action sequence for both genital attachment and detachment.

## Results

### Generating a *dsx* Split-GAL4 allele

To functionally identify sub-populations of *dsx* neurons with differing neurotransmitter profiles, we generated a novel *dsx* Split-GAL4 allele (*dsx*<sup>GAL4-DBD</sup>) by homologous recombination at the *dsx* locus (*Figure 1a,b*) (*Luan et al., 2006*). We validated the specificity of expression pattern of the *dsx* Split-GAL4 allele by pairing it with a pan-neuronal matching Split-GAL4 driver (*elav*<sup>VP16-AD</sup>) (*Luan et al., 2006*) (*Figure 1c,d*). *dsx/elav>GFP* flies replicated the expression pattern of the previously characterized *dsx*<sup>GAL4</sup> allele in the nervous system (*Figures 1d1–4, c1–4*, respectively) without exhibiting GFP expression in non-neural tissues (*Figures 1d5–6 vs. c5–6*). We then functionally validated *dsx*<sup>GAL4-DBD</sup> by silencing *dsx/elav* neurons with tetanus toxin light chain (TNT) (*Sweeney et al., 1995*), which blocks synaptic vesicle exocytosis, in both males and females (*Figure 1—figure supplement 1*). *dsx/elav>TNT* males and females reproduced the behavioral phenotypes of the previously characterized *dsx*<sup>GAL4</sup>>*TNT* males and females (*Rideout et al., 2010*). Specifically, *dsx/elav>TNT* males spent very little time courting wild type females (*Figure 1—figure supplement 1a*), and completely failed to copulate (*Figure 1—figure supplement 1b,c*) and were therefore behaviorally sterile (*Figure 1—figure supplement 1d*), while *dsx/elav>TNT* females were infertile (*Figure 1—figure supplement 1e*), unreceptive (*Figure 1—figure supplement 1f,g*), and exhibited no post-mating behavioral responses (*Figure 1—figure supplement 1h*).

### *dsx*/glutamatergic motor neurons control copulation

To identify *dsx* neurons involved in the motor control of the genitalia during copulation, we targeted *dsx* motor neurons using a Split-GAL4 insertion downstream of the *vesicular glutamate transporter* (*vGlut*) locus (*Gao et al., 2008*), derived from the OK371-GAL4 enhancer trap (*Mahr and Aberle, 2006*). Glutamate is the key excitatory neurotransmitter at the *Drosophila* neuromuscular junction (NMJ) (*Daniels et al., 2008*), and both OK371-GAL4 and *vGlut*<sup>OK371-dVP16-AD</sup> (referred to as *vGlut*<sup>dVP16-AD</sup> here onwards) (*Gao et al., 2008*) have been widely used to target motor neurons (*Karuppudurai et al., 2014*; *Ting et al., 2014*). We used *dsx*<sup>GAL4-DBD</sup>/*vGlut*<sup>dVP16-AD</sup> to drive nuclear

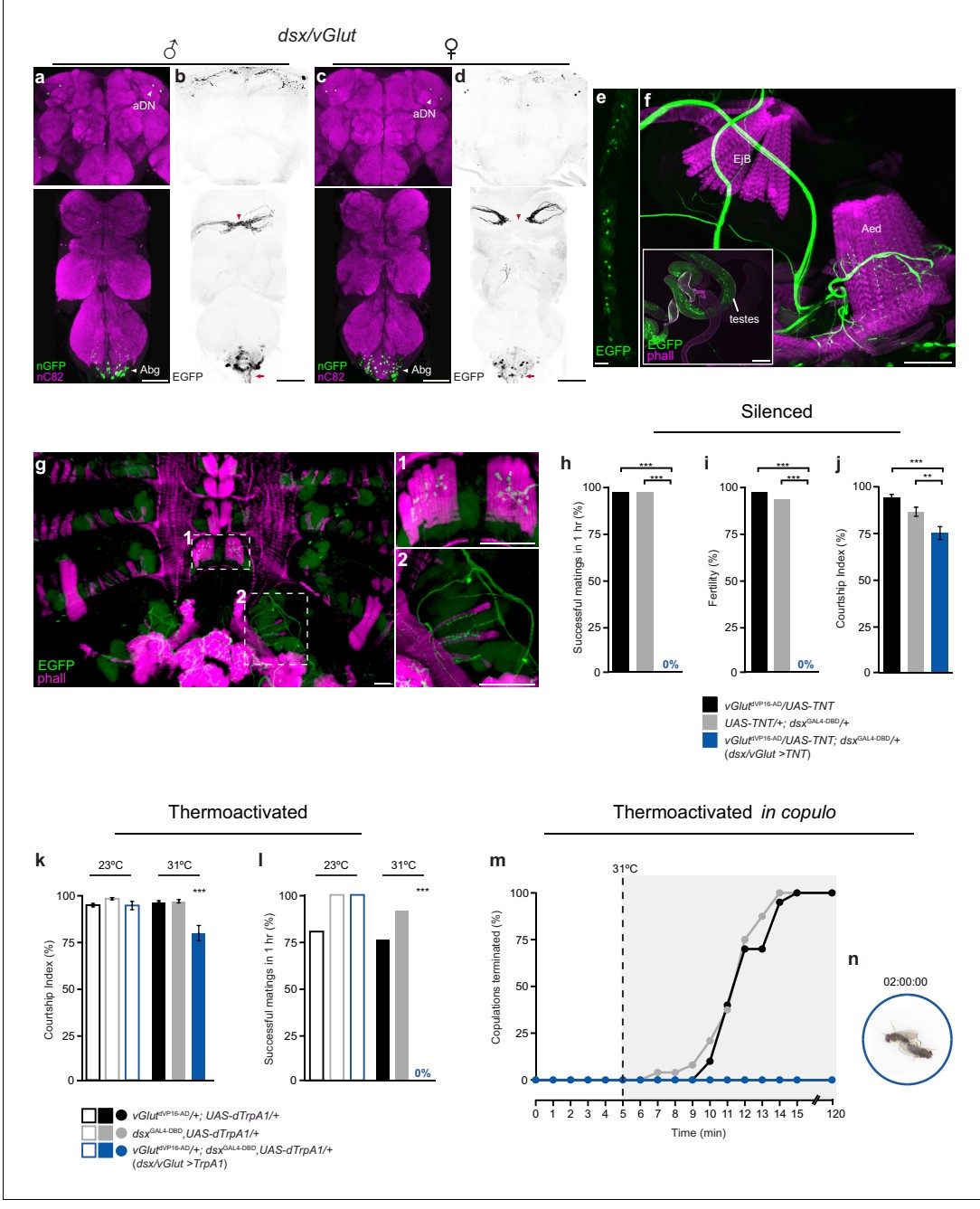

**Figure 2.** Sexually dimorphic *dsx*/glutamatergic neurons control genital coupling during copulation. (**a-d**) Sexually dimorphic expression of *dsx/vGlut* neurons in the brain (top) and VNC (bottom) of adult males (**a,b**) and females (**c,d**). (**a,c**) *dsx/vGlut* cell bodies visualized by *vGlut*^dVP16-AD^/*dsx*^GAL4-DBD^ driving *UAS-nGFP* in (**a**) male and (**c**) female CNSs. nGFP stained with anti-GFP (green); neuropil counterstained with anti-nC82 (magenta). (**b,d**) *dsx/vGlut* projection patterns visualized by *vGlut*^dVP16-AD^/*dsx*^GAL4-DBD^ driving *UAS-2XEGFP* in (**b**) male and (**d**) female CNSs. EGFP stained with anti-GFP (black); sexually dimorphic midline crossing (red arrowhead) and neurons of the Abg and their descending projections (red arrow) are shown. (**e**) *dsx/vGlut* driven EGFP expression in the T1 tarsi of the male foreleg. Projections from these neurons form the male-specific contralateral commissural bridge in the mesothoracic gangion of the male VNC (red arrowheads in bottom panels of **b,d**). (**f,g**) *dsx/vGlut* driven EGFP expression in the (**f**) internal reproductive system and (**g**) abdomen reveals motor neuron arborizations onto (**f**) muscles of the aedeagus and (**g**) dorsal and ventral muscles of the sixth abdominal segment. (**g1-2**) Higher magnification of ventral (**g1**) and dorsal (**g2**) longitudinal muscles of sixth abdominal segment showing *dsx/vGlut* motor neuron innervations and synaptic termini. EGFP stained with anti-GFP (green). Internal reproductive system

*Figure 2 continued on next page*

*Figure 2 continued*

and abdominal muscles counterstained with the F-actin specific antibody Phalloidin (phall; magenta). Detail of internal genitalia: testes, ejaculatory bulb (EjB), and aedeagus (Aed) indicated. Scale bar = 50 μm. (h-j) Effects of silencing *dsx/vGlut* neurons on male copulatory and courtship behaviors. (h) Percentage of successful matings in 1 hr (n = 24–30). (i) Male fertility (n = 30). (j) Courtship index (mean ± S.E.M.; n = 24–30). Genotypes indicate males. See also *Video 1*. (k,l) Effects of thermoactivating *dsx/vGlut* neurons on male courtship and copulatory behaviors. (k) Courtship index (mean ± S.E.M.; n = 20–30). (l) Percentage of successful matings in 1 hr (n = 20–30). Statistical comparisons of the experimental genotype at 31°C in (k-l) were made against the same genotype at 23°C and all control genotypes at 31°C. Genotypes indicate males. (m) Effects of thermoactivating male *dsx/vGlut* neurons 5 min into copulation. Percentage of copulations terminated over a 2 hr period is graphed (n = 22–24). (n) Video still showing 'stuck' *dsx/vGlut>TrpA1* male at the end of the 2 hr observation period. See also *Video 2*. (h-l) **p<0.001, ***p<0.0001 by Fisher exact test (h,i,k) or Kruskal-Wallis and Dunn's test (j,l).
The following figure supplement is available for figure 2:

**Figure supplement 1.** Characterisation of *dsx*/glutamatergic neurons in the adult male CNS.

GFP to count cell numbers and cytoplasmic GFP to visualize neuronal projections (*Figure 2a–g*). *dsx/vGlut* neurons were found in three anatomically distinct regions in both males and females: neurons in the Abg, forelegs, and *dsx*-aDN neurons in the brain (*Figure 2a–e*; *Table 1*). No cells were labeled elsewhere in the PNS, or non-neural tissues. Co-labeling with anti-dvGlut (*Mahr and Aberle, 2006*) antibody confirmed the glutamatergic identity of $dsx^{GAL4-DBD}/vGlut^{dVP16-AD}$ neurons (*Figure 2—figure supplement 1a–c*).

The largest sub-population of *dsx/vGlut* neurons is that of the Abg (with ~80 in males and ~100 in females; *Figure 2a,c*; *Table 1*). These project from the VNC via the abdominal nerve trunk (*Figure 2b,d*) and arborize onto muscles of the genitalia of both sexes. In males this includes the muscles controlling protraction and retraction of the aedeagus (*Figure 2f*) as well as the most distal (A6) longitudinal muscles of the ventral and dorsal abdomen (*Figure 2g*). We further confirmed that *dsx/vGlut* neurons encompass all of the motor neurons that innervate all phallic and periphallic musclulature (*Figure 2—figure supplement 1f*). In females, these neurons innervate muscles of

**Table 1.** Cell counts for *dsx*-intersected neurons in male and female adult CNS. Male and female *dsx/elav*, *dsx/vGlut* and *dsx/Gad1* cell counts are listed in black. Subsets of neurons that co-express Fru[M] in males are listed in italics.

| dsx neuronal clusters | dsx/elav | | dsx/vGlut | | dsx/Gad1 | |
|---|---|---|---|---|---|---|
| | Male | Female | Male | Female | Male | Female |
| Brain | | | | | | |
| pC1* | 52.8 ± 4.1 (12) | 8.3 ± 1.6 (12) | 0 ± 0 (12) | 0 ± 0 (12) | 0 ± 0 (12) | 0 ± 0 (12) |
| pC2* | 78.3 ± 4.8 (12) | 14.2 ± 1.5 (12) | 0 ± 0 (12) | 0 ± 0 (12) | 1.0 ± 0(12) *0.9 ± 0.3 (10)* | 0 ± 0 (12) |
| pC3* | 13.8 ± 0.9 (12) | 8.0 ± 1.0 (12) | 0 ± 0 (12) | 0 ± 0 (12) | 3.5 ± 0.5 (12) *0.5 ± 0.5 (10)* | 3.0 ± 0 (12) |
| aDN* | 2.0 ± 0 (12) | 2.0 ± 0 (12) | 1.9 ± 0.3 (12) *0 ± 0 (10)* | 2.0 ± 0 (12) | 0 ± 0 (12) | 0 ± 0 (12) |
| SN* | 1.0 ± 0 (12) | n.a. | 0 ± 0 (12) | n.a. | 0 ± 0 (12) | n.a. |
| Ventral Nerve Cord | | | | | | |
| TN1* | 23.0 ± 1.5 (12) | n.a. | 0 ± 0 (12) | n.a. | 0 ± 0 (12) | n.a. |
| TN2* | 7.9 ± 0.3 (12) | n.a. | 0 ± 0 (12) | n.a. | 0 ± 0 (12) | n.a. |
| Abg† | 275.0 ± 21.7 (10) | 314.8 ± 18.9 (10) | 79.8 ± 2.3 (10) *7.4 ± 3.1 (10)* | 101.8 ± 6.7 (10) | 151.2 ± 3.8 (10) *30.0 ± 4.8 (10)* | 213.1 ± 2.1 (10) |

*Neuronal cluster away from CNS midline. Count represents one cluster per hemisegment of the CNS.
†Neuronal cluster spans the CNS midline. Count given is for the entire Abg. Counts represent mean ± S.D. n's listed in parentheses.

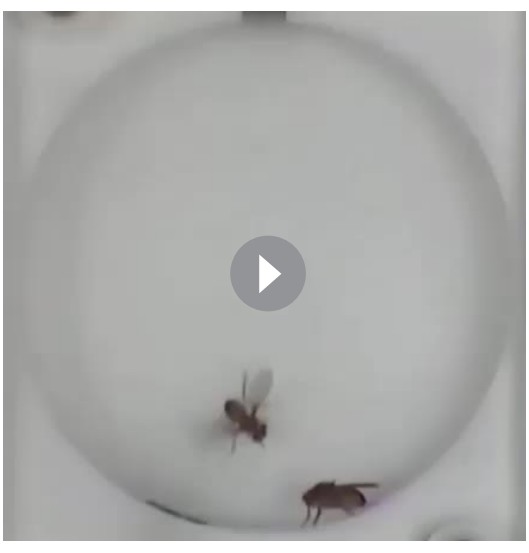

**Video 1.** Inhibition of *dsx/vGlut* neurons in males blocks genital coupling and the initiation of copulation. This movie shows a *dsx/vGlut>TNT* male failing to achieve genital coupling and initiate copulation with a wild-type female. These males do however display the normal complement of courtship behaviors.

uterus, spermathecal and parovarian ducts (data not shown). No additional muscular innervations were observed. Neurons in the foreleg (*Figure 2e*) project to the prothoracic ganglion and cross the midline in males, but not females (*Figure 2b,d*), which is typical of foreleg gustatory receptor neurons (GRNs) (*Possidente and Murphey, 1989*; *Mellert et al., 2010*; *Rideout et al., 2010*). The two neurons of the *dsx*-aDN cluster project locally within the dorsal brain and extend to the superior medial protocerebrum (SEM) in both sexes (*Figure 2a–d*; *Table 1*). To examine a potential role for *fru* in specifying the sexual identity of *dsx/vGlut* neurons, we co-stained samples with Fru$^M$ antibody (*Figure 2—figure supplement 1a,d,e*). Interestingly, none of the *dsx/vGlut*-aDN and only ~10% of *dsx/vGlut*-Abg neurons co-expressed Fru$^M$ (*Figure 2—figure supplement 1d,e* respectively; *Table 1*). These *fru/dsx/vGlut*-Abg neurons likely include the motor neurons that innervate the dorsal and ventral muscles of the sixth abdominal segment (*Nojima et al., 2010*).

We tested the role of *dsx/vGlut* motor neurons in copulation by silencing their activity with TNT. *dsx/vGlut>TNT* males completely failed to achieve genital coupling (*Figure 2h*; *Video 1*). Even after seven days in the presence of several virgin females, *dsx/vGlut>TNT* males produced no progeny (*Figure 2i*). These males also spent less time courting target females (*Figure 2j*), although they displayed the normal complement of courtship behaviors, including attempting to copulate (*Video 1*). We conclude that *dsx/vGlut* neurons are necessary for successful genital coupling.

Copulation requires motor coordination between the external genitalia and the copulatory organ to facilitate genital attachment. We therefore tested whether simultaneous contraction of these organs prior to or during copulation, by artificial activation of all *dsx/vGlut* neurons, would prevent genital coupling. We expressed the *Drosophila* heat-activated cation channel dTrpA1 (*Hamada et al., 2008*) in *dsx/vGlut* neurons and examined the effects of *dsx/vGlut* thermoactivation prior to copulation. Pairs of *dsx/vGlut>dTrpA1* experimental males and wild-type females were heated to 31°C 10 min prior to and throughout a 1 hr observation period (*Figure 2k, l*). Thermoactivated *dsx/vGlut* males displayed less overall courtship towards target females (*Figure 2k*); as with *dsx/vGlut>TNT* males, these males displayed all courtship steps, but never successfully copulated (*Figure 2l*). These results indicate that disrupting the activity of *dsx/vGlut* neurons pre-copulation, by either complete silencing or activation, perturbs the motor events that are necessary for a male to attach to a female and initiate copulation (*Figure 2h,l*).

**Video 2.** Activation of *dsx/vGlut* neurons in copulating males blocks genital uncoupling and the termination of copulation. This movie shows three *dsx/vGlut>TrpA1* males that remain attached to their wild-type female mating partners via their genitals after ~2 hr of *in copulo* thermal activation (31°C).

Insemination occurs within the first ~8 min of copulation (*Gilchrist and Partridge, 2000*). During this critical time, males resist being interrupted by any stressful stimuli, displaying 'copulation persistence' (*Crickmore and Vosshall, 2013*). To test the role of *dsx/vGlut* neurons during copulation, we activated *dsx/vGlut* neurons 5 min into copulation, when fertilization is not complete, and 'copulation persistence' is at its peak (*Crickmore and Vosshall, 2013*) (*Figure 2m,n*; *Video 2*). Compellingly, throughout a 2 hr observation period, thermoactivation of *dsx/vGlut* neurons *in copulo* prevented the male from detaching from the female and terminating copulation, which normally occurs after 10–15 min (*Figure 2m,n*; *Video 2*). Extended thermoactivation of *dsx/vGlut* neurons *in copulo* did not disrupt sperm transfer, as all matings were fertile. Interestingly, stimulation of *dsx/vGlut* neurons did not impair the timing of copulation drive; after approximately 15 min, *dsx/vGlut>dTrpA1* males attempted to uncouple from the female by dismounting and/or kicking the female genitalia with their hind legs.

Which of the anatomically distinct *dsx/vGlut* neurons are responsible for this phenotype? Amputation of the forelegs results in indiscriminate courtship towards con- and allo-specific males and females (*Fan et al., 2013*), but does not impinge upon copulatory behavior(s), indicating that *dsx/vGlut* neurons in the foreleg may effectively be ruled out of this particular motor circuit. Furthermore, decapitation of thermoactived *dsx/vGlut>dTrpA1* males *in copulo* did not eliminate the 'stuck' phenotype (*Figure 2—figure supplement 1g*). These results demonstrate that *dsx/vGlut*-Abg neurons comprise the key motor neurons that control genital coupling during a copulation event.

## *dsx*/GABAergic neurons in the abdominal ganglion control the termination of copulation

To identify *dsx* inhibitory neurons we focused on GABA, the major inhibitory neurotransmitter in insects (*Jackson et al., 1990*). As GABA biosynthesis requires glutamic acid decarboxylase 1 (Gad1) (*Erlander and Tobin, 1991*), we exploited the intersecting expression domains of *dsx* and *Gad1* using *dsx*[GAL4-DBD] and a *Gad1* hemi-drivers (*Gad1*[p65-AD]) (*Diao et al., 2015*). We observed sexually dimorphic groups of *dsx/Gad1* neurons in both the Abg and brain (*Figure 3a–d*; *Table 1*). There was no expression in the PNS or non-neural tissues. Co-labeling with anti-GABA antibody confirmed that neurons labeled by *dsx*[GAL4-DBD]/*Gad1*[p65-AD] are indeed GABAergic (*Figure 3—figure supplement 1a–c*).

In the Abg, we observed ~150 *dsx/Gad1* neurons in males and ~210 in females, or over 50% and 65% of all *dsx* Abg neurons, respectively (*Table 1*). *dsx/Gad1*-Abg neurons arborize locally within the Abg, except for two neurons that project to the mesothoracic and prothoracic ganglia in males only (*Figure 3b,d*). In the central brain, *dsx*[GAL4-DBD]/*Gad1*[p65-AD] labeled a small subset of *dsx*-pC2 and -pC3 (*Rideout et al., 2010*) neurons in males, and an equivalent subset of *dsx*-pC3 neurons in female that project to the SEM and subesophageal zone (SEZ; *Figure 3a–d*, see cell counts in *Table 1*). To determine whether *dsx/Gad1* neurons express *fru*, we co-labeled male CNSs with Fru[M] antibody (*Figure 3—figure supplement 1d,e*). We observed *fru* expression in the *dsx/Gad1*-pC2 neurons, in one out of the three *dsx/Gad1*-pC3 neurons per brain hemisphere, and in ~20% of *dsx/*GABAergic neurons in the Abg (*Figure 3—figure supplement 1d,e*; *Table 1*).

We tested the role of *dsx/Gad1* neurons in copulation by silencing their activity with TNT (*Figure 3e–i*). *dsx/Gad1>TNT* males displayed relatively normal levels of courtship towards target females and displayed all courtship steps, including abdominal bending and attempted copulation (*Figure 3e*). However, less than 20% of males successfully copulated during a 1 hr observation period (*Figure 3f*). The males that successfully copulated took significantly longer than controls (*Figure 3g*). Approximately ~ 80% of these males could not uncouple from the female (*Figure 3h*; *Video 3*), exhibiting a similar 'stuck' phenotype to that of *dsx/vGlut* males thermoactivated *in copulo* (*Figure 2m,n*; *Video 2*). Sperm transfer appeared normal, as all of these matings produced progeny (data not shown). After one week in the presence of several virgin females, only ~60% of *dsx/Gad1>TNT* males produced progeny (*Figure 3i*). The timing of copulatory motivation appeared to be intact, because after approximately 15 min, *dsx/Gad1>TNT* males dismounted and attempted (but failed) to detach. We conclude that *dsx/Gad1* neurons functionally oppose *dsx/vGlut*-Abg neurons: *dsx/vGlut*-Abg neurons promote genital coupling while *dsx/Gad1* neurons promote genital uncoupling.

We hypothesized that if *dsx/Gad1* neurons inhibit *dsx/vGlut* motor neurons, artificial activation of these neurons pre-copulation should prevent males from achieving genital attachment. To test this,

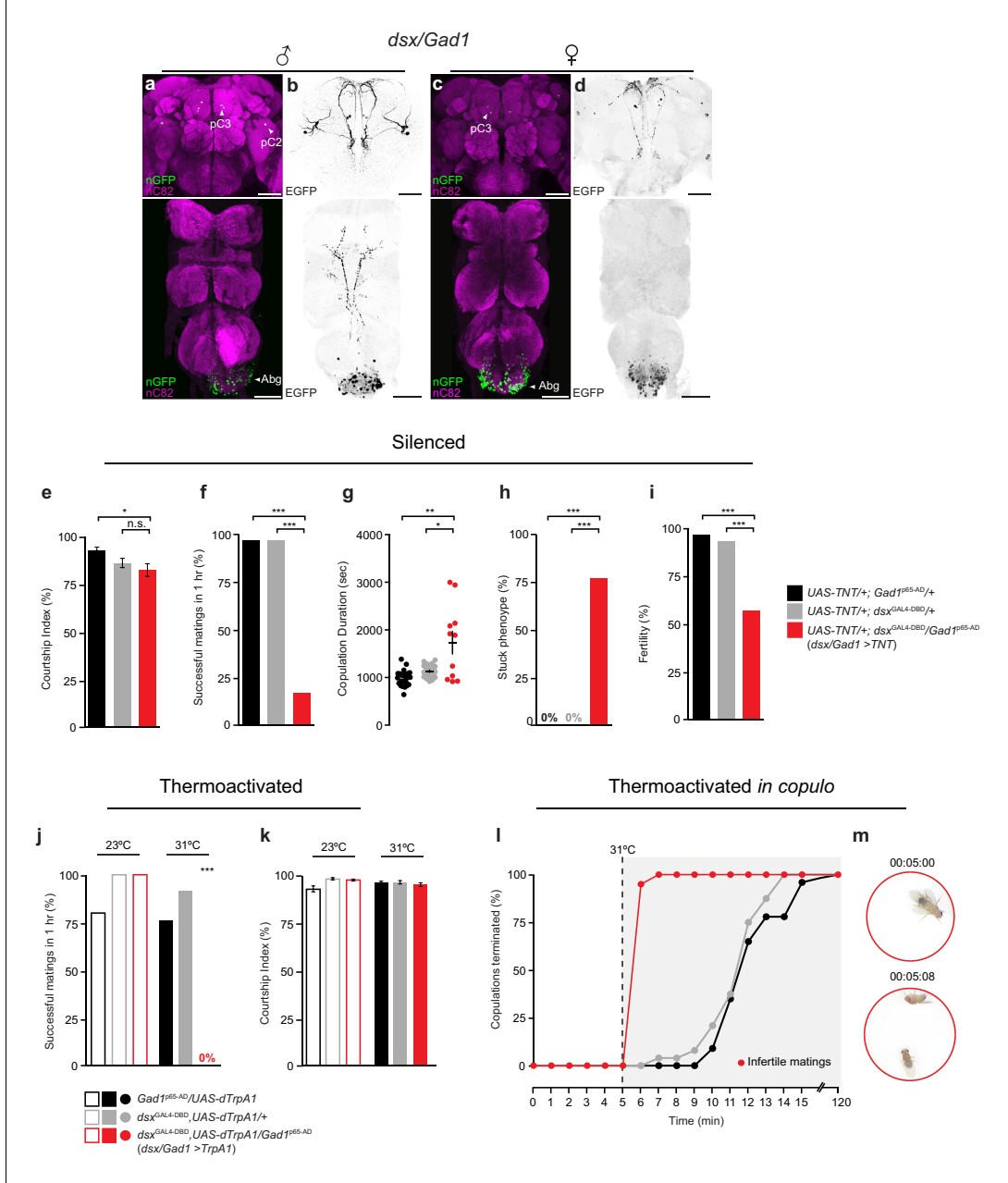

**Figure 3.** Sexually dimorphic *dsx*/GABAergic neurons control genital uncoupling during copulation. (**a-d**) Sexually dimorphic expression of intersected *dsx/Gad1* neurons in the brain (top) and VNC (bottom) of adult males (**a,b**) and females (**c,d**). (**a,c**) *dsx/Gad1* cell bodies visualized by *dsx*GAL4-DBD/*Gad1*p65-AD driving *UAS-nGFP* in (**a**) male and (**c**) female CNSs. nGFP stained with anti-GFP (green); neuropil counterstained with anti-nC82 (magenta). (**b,d**) *dsx/Gad1* projection patterns visualized by *dsx*GAL4-DBD/*Gad1*p65-AD driving *UAS-2XEGFP* in (**b**) male and (**d**) female CNSs. EGFP stained with anti-GFP (black). Scale bar = 50 μm. (**e-i**) Effects of silencing *dsx/Gad1* neurons on male copulatory and courtship behaviors. (**e**) Courtship index (mean ± S.E.M.; n = 24–30). (**f**) Percentage of successful matings in 1 hr (n = 24–30). (**g**) Copulation duration (n = 12–24). (**h**) Percentage of males displaying 'stuck' phenotype (n = 12–24). (**i**) Male fertility (n = 30). Genotypes indicate males. See also **Video 3**. (**j,k**) Effects of thermoactivating *dsx/Gad1* neurons on male courtship and copulatory behaviors. (**j**) Percentage of successful mating's in 1 hr (n = 20–30). (**k**) Courtship index (mean ± S.E.M.; n = 20–30). Statistical comparisons of the experimental genotype at 31°C in (**j,k**) were made against the same genotype at 23°C and all control genotypes at 31°C. Genotypes indicate males. (**l**) Effects of thermoactivating male *dsx/Gad1* neurons 5 min into copulation. Percentage of copulations terminated over a 2 hr period is graphed (n = 20–24). (**m**) Video stills showing that activation of *dsx/Gad1* >*TrpA1* neurons *in copulo* results in an almost immediate termination of copulation. Top

*Figure 3 continued on next page*

*Figure 3 continued*
panel shows *dsx/Gad1 >TrpA1* male and wild type female mating 5 min into copulation at the point of shifting the temperature to 31°C. Bottom panel shows the same mating pair 8 s later, at which time the male has terminated copulation. See also *Video 4*. (e-k) n.s. = not significant, *p<0.05, *p<0.05, **p<0.001, ***p<0.0001 by Fisher exact test (f,h-j) or Kruskal-Wallis and Dunn's test (e,g,k).

The following figure supplements are available for figure 3:

**Figure supplement 1.** Characterisation of *dsx*/GABAergic neurons in the adult male CNS.

**Figure supplement 2.** *dsx*/GABAergic neurons in the male brain do not specify male copulatory behaviors, but instead mediate male courtship.

we expressed dTrpA1 in *dsx/Gad1* neurons and examined the effects of *dsx/Gad1* thermoactivation on courtship and copulatory behaviors. Thermoactivating *dsx/Gad1* neurons in these males completely blocked genital attachment in the 1 hr observation period (*Figure 3j*). The males' inability to achieve genital coupling was not a result of impaired courtship, as they spent normal amounts of time courting (*Figure 3k*) and displayed all courtship behaviors, including vigorous, but failed, attempts to copulate. Specifically, *dsx/Gad1>dTrpA1* males attempted to copulate 34 ± 3 times within the first 10 min of courtship, while controls attempted to copulate 7 ± 1 times prior to a successful copulation (n = 10). As blocking *dsx/Gad1* neurons prolongs copulation and activating them prevents it, we conclude that *dsx/Gad1* neurons promote genital uncoupling. Silencing *dsx/Gad1* neurons and activating *dsx/vGlut* neurons both disrupt genital coupling, suggesting that *dsx/Gad1* neurons may selectively inhibit *dsx/vGlut* neurons to create the precise pattern of *dsx/vGlut* activity required for copulation.

Artificial activation of *dsx/Gad1* neurons *in copulo* should therefore inhibit genital coupling and result in premature termination of copulation. Compellingly, ~90% of *dsx/Gad1>dTrpA1* males dismounted females and terminated copulations within 1 min of thermoactivation (*Figure 3l,m*; *Video 4*); this was significantly shorter than the termination times of control males (*Figure 3l*). Given that the majority of *dsx/Gad1>dTrpA1* males spent no more ~6 min *in copulo*, much less than the ~8 min requirement for fertile matings (*Figure 3l*) (*Gilchrist and Partridge, 2000*), we checked the fertility of each mating. Matings truncated by thermoactivation never produced any progeny (*Figure 3l*). These results demonstrate that *dsx/Gad1* neurons are sufficient to induce genital uncoupling and terminate copulation. Importantly, the opposing phenotypes that result from thermoactivating *dsx/Gad1* and *dsx/vGlut* neurons *in copulo* (premature termination and perpetual copulation, respectively) demonstrate that *dsx/Gad1* neurons functionally oppose *dsx/vGlut* neurons to inhibit genital coupling for the termination of copulation.

To determine whether *dsx/Gad1* neurons in the brain or in the Abg caused the genital uncoupling phenotype, we combined the *dsx*GAL4-DBD/*Gad1*p65-AD driver with the brain-specific flippase *Otd-nls:FLPo* (*Otd*FLP) (*Asahina et al., 2014*) and *UAS*-driven reporters or effectors with FRT-flanked stop cassettes (*Figure 3—figure supplement 2*). This intersectional combination enabled selective expression of reporters and effectors in *dsx/Gad1* neurons only in the brain (*Figure 3—*

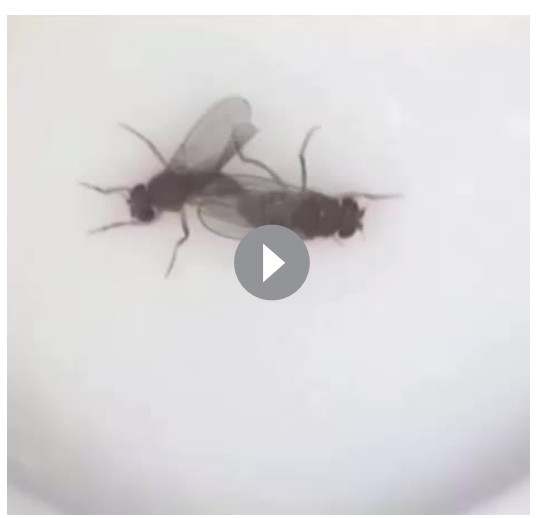

**Video 3.** Inhibition of *dsx/Gad1* neurons in males blocks genital uncoupling and the termination of copulation. This movie shows a *dsx/Gad1>TNT* male displaying the distinctive 'stuck' behavior, whereby he has dismounted the wild-type female but remains attached via his genitals for prolonged periods of time.

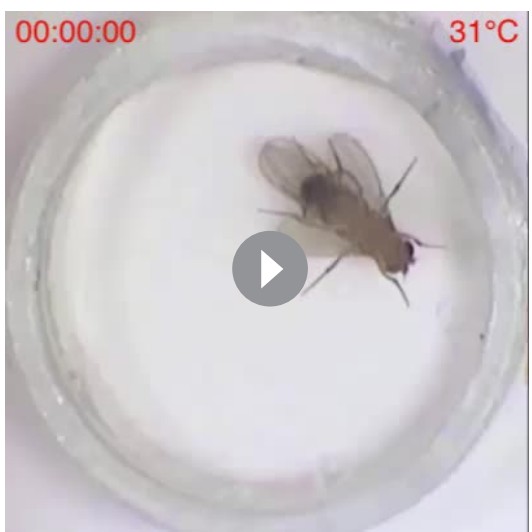

**Video 4.** Activation of *dsx/Gad1* neurons in copulating males elicits genital uncoupling and the termination of copulation. This movie shows a *dsx/vGlut>TrpA1* male and wild-type female copulating pair that have been shifted to 31°C 5 min into copulation. Thermal activation of male *dsx/vGlut* neurons in this manner results in the near-immediate termination of copulation by *dsx/Gad1>TrpA1* males.

*figure supplement 2*). *Otd/dsx/Gad1>TNT* males with silenced *dsx/Gad1* brain neurons spent less time courting females than controls (*Figure 3—figure supplement 2c*) but exhibited no copulatory defects (*Figure 3—figure supplement 2d,e*). In addition, thermoactivation of *Otd/dsx/Gad1>dTrpA1* males, both pre-copulation and *in copulo*, did not impair courtship or copulation (*Figure 3—figure supplement 2f–i*). Comparing these data to the phenotypes of *dsx/Gad1>TNT* males (*Figure 3e–i*) demonstrate that the observed copulation defects stem from the *dsx/Gad1*-Abg neurons.

## *dsx/vGlut* neurons are poised for *dsx/Gad1* inhibition

The opposing functions of *dsx/Gad1* and *dsx/vGlut* neurons suggest that *dsx/Gad1* neurons may inhibit *dsx/vGlut* neurons. To test this hypothesis anatomically, we expressed the presynaptic reporter *UAS-nSyb-GFP* (*Estes et al., 2000*) in *dsx/Gad1* neurons (*Figures 4a1-a1'*), and the dendritic marker 'DenMark' (*UAS-DenMark*) (*Nicolaï et al., 2010*) in *dsx/vGlut* neurons (*Figures 4a2-a2'*) independently, and registered the images to a standardized template VNC (*Jefferis et al., 2007*; *Cachero et al., 2010*; *Ostrovsky et al., 2013*) (*Figure 4a–a'*). Computational alignment revealed that *dsx/vGlut* dendrites overlap with *dsx/Gad1* presynaptic boutons (*Figure 4a–a'*; *Video 5*), indicating that *dsx/Gad1* neurons are anatomically positioned to inhibit *dsx/vGlut* neurons.

To test this hypothesis functionally, we asked whether knocking down GABA receptors in *dsx/vGlut* neurons would recapitulate the *dsx/Gad1>TNT* phenotype. We behaviorally screened all five GABA receptor subunits by RNAi knockdown in *dsx/vGlut* neurons (*Figure 4b* and *Figure 4—figure supplement 1a–c*). *dsx/vGlut>GABA_B-R2-RNAi* knockdown resulted in a significant defect in copulation termination, with ~15% of males displaying a 'stuck' phenotype (*Figure 4b*). The reduced strength of this phenotype compared to *dsx/Gad1>TNT* flies may be due to incomplete knockdown of the $GABA_B$ receptor by RNAi (*Dietzl et al., 2007*; *Lin et al., 2014*). These results suggest that *dsx/Gad1* neurons help terminate copulation at least partially through inhibiting *dsx/vGlut* neurons via metabotropic $GABA_B$ receptors. Taken together, our data support the notion that *dsx/vGlut* neurons are poised for *dsx/Gad1* inhibition.

## *dsx* genital neurons relay sensory information to *dsx* abdominal neurons

Mechanosensory sensilla on the male genital claspers and lateral plates have species-specific roles in establishing correct mating posture and genital coupling during the initial stages of copulation (*Acebes et al., 2003*; *Jagadeeshan and Singh, 2006*). We postulated that sensory information from the genitalia could provide feedback to neurons controlling copulation during genital attachment and copulation. To identify sensory neurons, we examined the peripheral nervous system (PNS) in *dsx/elav>GFP* flies (*Figure 1d*). We uncovered novel patterns of *dsx*-expression in the PNS (*Figure 5*). As expected, we identified sexually dimorphic patterns of *dsx*-expression in several bilateral clusters of mechanosensory neurons of the male and female adult terminalia (*Figure 5b1,c1*), in addition to the foreleg and labellum of both sexes (*Figure 5b2–3, c2–3*, respectively). In males, *dsx* neurons in the terminalia are associated with bristles of the clasper teeth, lateral plates and anal

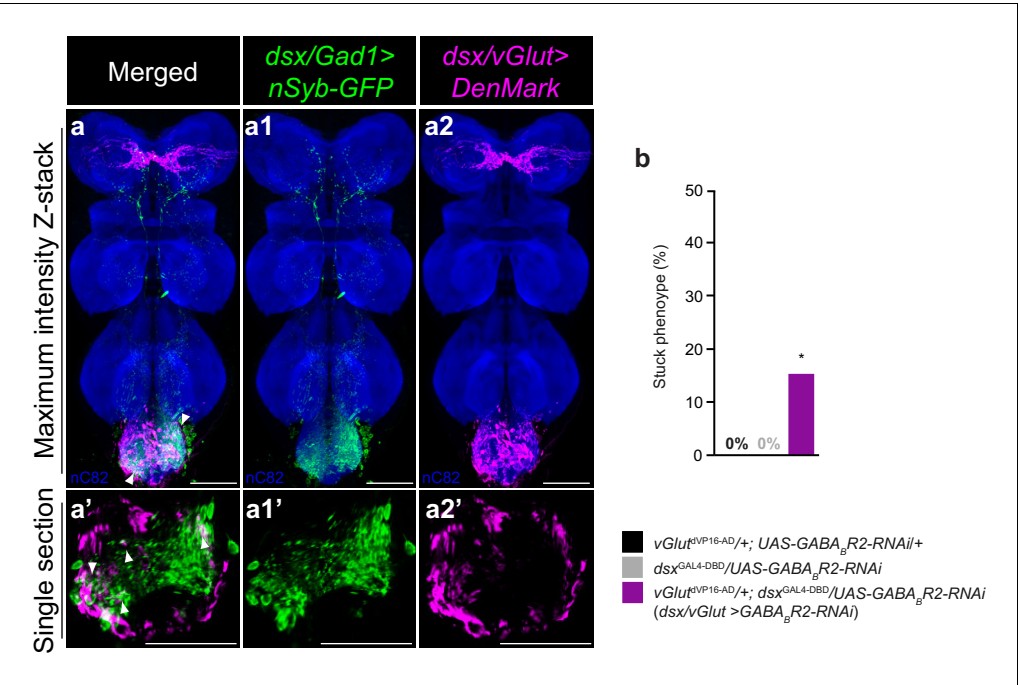

**Figure 4.** *dsx*/glutamatergic neurons are poised for *dsx*/GABAergic inhibition. (a–a2) Overlay of expression of (a1) *dsx/Gad1* presynaptic boutons and *dsx/vGlut* dendrites on standardized template VNC. (a1) *dsx/Gad1* presynaptic boutons visualized by *dsx*[GAL4-DBD]/*Gad1*[P65-AD] driving *UAS-nSyb::GFP* in male VNC stained with anti-GFP (green). (a2) *dsx/vGlut* dendrites visualized by *vGlut*[dVP16-AD]/*dsx*[GAL4-DBD] driving *UAS-DenMark* in male VNC stained with anti-DsRed (magenta); neuropil counterstained with anti-nC82 (blue). (a–a2) Maximum intensity z-stacks and (a'–a2') single section images. Solid arrowheads point at regions of close proximity. Scale bar = 50 μm. See also *Video 5*. (b) Effects of knocking down GABA$_B$-R2 receptor by RNAi in *dsx/vGlut* neurons on male copulatory behavior. Percentage of males displaying a 'stuck' phenotype is graphed (n = 20–30). *p<0.05 by Fisher exact test.

The following figure supplement is available for figure 4:

**Figure supplement 1.** Effects of knocking down GABA receptor subunits in *dsx*/glutamatergic neurons on copulatory behaviors.

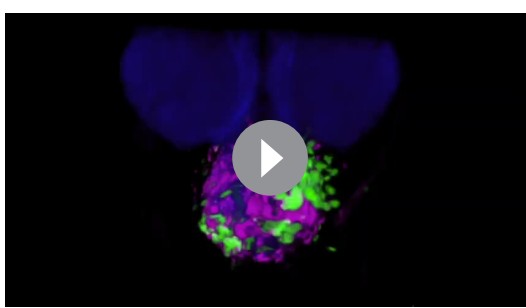

**Video 5.** *dsx/Gad1* presynaptic boutons are in close proximity to *dsx/vGlut* dendrites. This movie shows the 3D reconstruction of a template abdominal ganglion showing an overlay of *dsx/vGlut* dendrites (magenta) and *dsx/Gad1* presynaptic boutons (green).

plates (*Figure 5b1*), and are cholinergic (*Yasuyama and Salvaterra, 1999*). In females, they are associated with bristles of the anal (not shown) and vaginal plates (*Figure 5c1*).

We asked whether these neurons form arborizations that overlap with *dsx/Gad1* or *dsx/vGlut* Abg neurons and therefore might provide input to them. Lacking drivers that specifically label mechanosensory neurons of the genitalia, we used the lipophilic carbocyanide DiD to retrograde label axons of mechanosensory neurons of the claspers, lateral plates, and hypandrium. Consistent with previous findings, we observed sexually dimorphic axonal projections, which show evidence of somatotopic organization in the Abg (*Figure 6a–g*; *Videos 6–8*) (*Taylor, 1989*). Interestingly however, by incubating the dye for 10 days, we also identified for the first time a single afferent axon (per

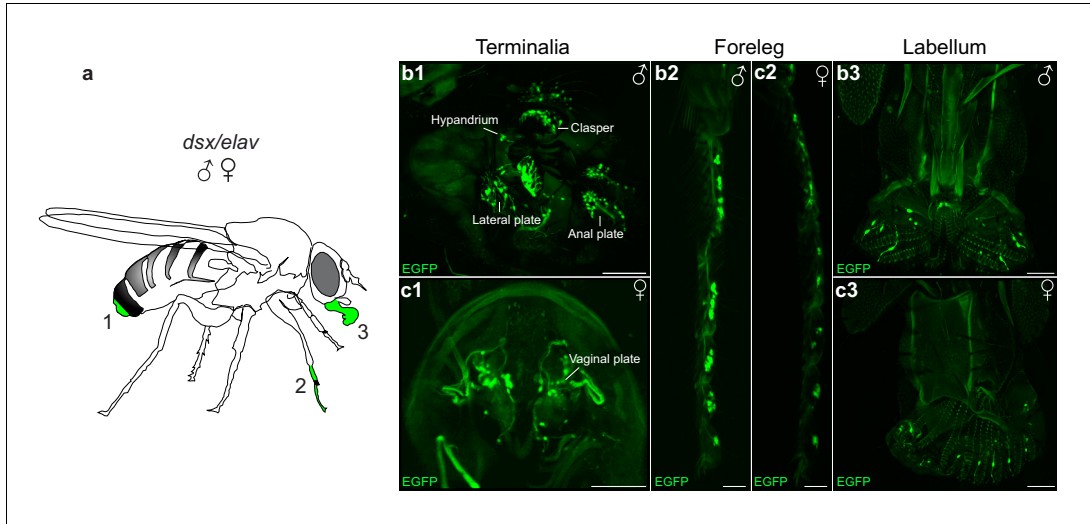

**Figure 5.** Novel patterns of *dsx* expression in the male and female peripheral nervous system. (a) Cartoon of adult fly depicting regions of *dsx*-expression in peripheral sense organs in males and females; **a1**: terminalia, **a2**: foreleg and **a3**: labellum of the mouthparts. (b,c) Sexually dimorphic *dsx*^GAL4-DBD^/*elav*^VP16-AD^ expression in peripheral sense organs in male and female adult flies. *dsx/elav* EGFP expression in bristle sensory neurons of the (**b1**) male clasper teeth, lateral plates, hypandrium and anal plates of the male terminalia and (**c1**) female vaginal plates of the female terminalia. *dsx/elav* EGFP expression in sensory neruons of the T1 tarsus of the foreleg in both males (**b2**) and females (**c2**). *dsx/elav* EGFP expression in sensory neruons of the labellum in both males (**b3**) and female (**c3**). EGFP is shown in green. Scale bar = 50 µm.

hemisegment) from the claspers that arborizes contralaterally within the Abg and traverses the entire VNC to ultimately terminate in the SEZ of the brain (*Figure 6h*). We then performed these dye fills in flies expressing the dendritic marker 'DenMark' in either *dsx/Gad1* or *dsx/vGlut* neurons (*Figure 6i, i'*). Native expression of DenMark in *dsx/Gad1* neurons was too weak to visualize dendritic boutons of *dsx/Gad1* neurons and establish a definitive relationship with genital neurons. However, *dsx/ Gad1* neurons were observed to have prolific dendrites in regions of the Abg occupied by genital sensory terminals (*Figure 4—figure supplement 1d*). Interestingly, projections of clasper, lateral plate and hypandrium neurons clearly interdigitated with the DenMark signal in *dsx/vGlut* in the Abg (*Figure 6i,i'*), suggesting that *dsx/vGlut* excitatory Abg neurons are poised to receive sensory information from the genitalia.

## Mechanosensory neurons of the genitalia feedback to *dsx* abdominal neurons

To establish direct functional connectivity between *dsx/vGlut* or *dsx/Gad1* neurons and mechanosensory neurons of the genitalia we expressed the calcium indicator GCaMP6m (*UAS-GCaMP*) (*Chen et al., 2013*) in *dsx/vGlut* or *dsx/Gad1* neurons and imaged responses evoked by genital stimulation (*Figure 7*). We stimulated the genital sensory neurons using a minutien pin attached to a micromanipulator. As a negative control, we stimulated segment A5 of the abdomen. Both *dsx/ vGlut* and *dsx/Gad1* Abg neurons responded strongly to genital, but not abdominal, touch (*Figure 7c–j*, *Videos 9,10*). Only a subset of neurons responded; activity maps show that not all areas of the neuropil respond significantly (*Figure 7d,g*; *Videos 9,10*), which may be due to regional mechanosensory stimulation. Nonetheless, these results show that sensory neurons of the genitalia functionally connect to *dsx/vGlut* and *dsx/Gad1* Abg neurons, and suggest that sensory feedback during copulation alters the activity of an Abg circuit that controls genital coupling and copulation.

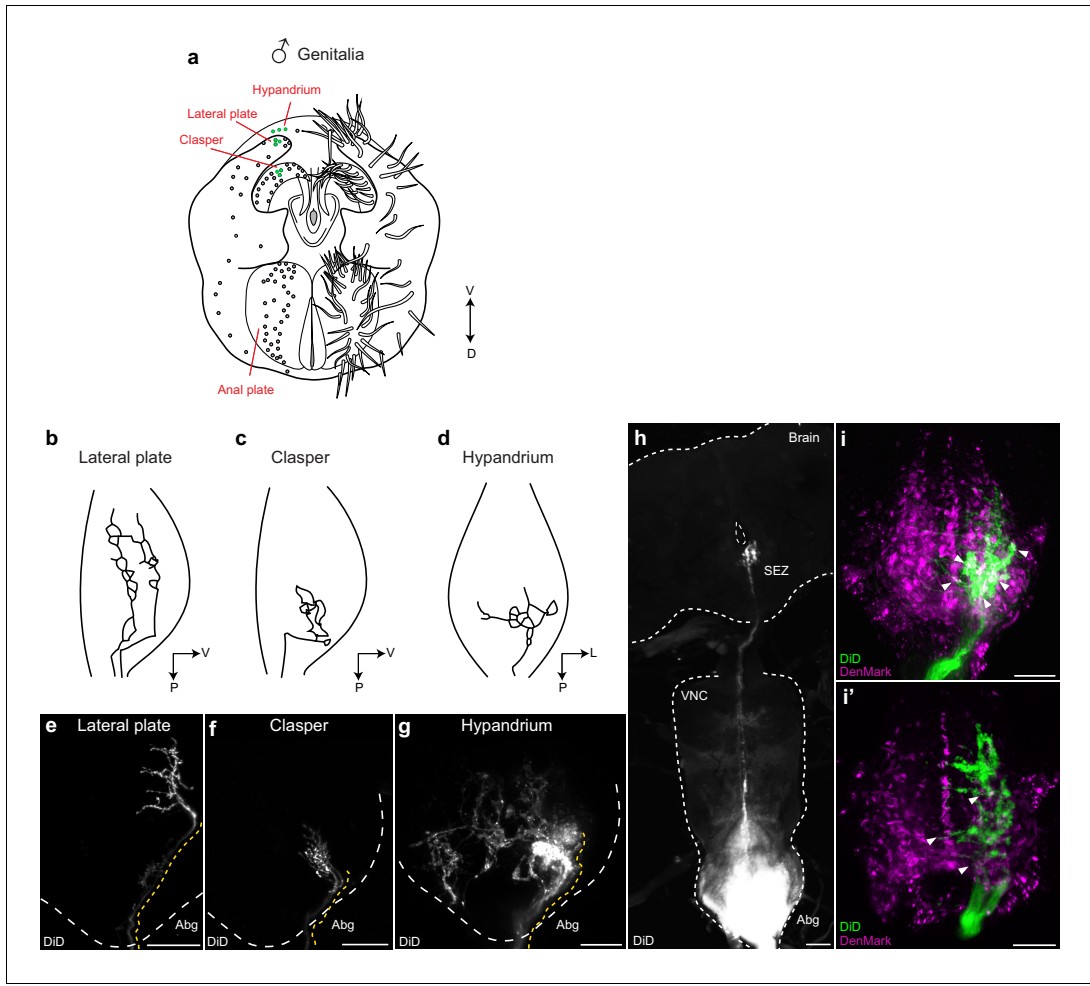

**Figure 6.** Mechanosensory neurons of the genitalia arborize onto the Abg and brain, and interdigitate with glutamatergic *dsx* motor neurons in the Abg. (a) Schematic of male terminalia depicting bristles (right) and bristle topography (dots on left). Dye-filled bristle topography in shown with green dots. (b-d) Schematic of representative lateral plate (b), clasper (c), and hypandrium (d) arborizations in the abdominal ganglion (Abg) of male flies, as previously described (*Taylor, 1989*). (e-g) Representative images showing topographically distinctive patterns of dye-filled genital neuron arborizations in the male Abg. Maximum intensity z-projections of confocal stacks showing unilateral arborizations of (e) lateral plate, (f) clasper, and (g) hypandrium neurons in the male Abg, which reiterate previously described arborizations (*Taylor, 1989*). Afferent projections from lateral plate and clasper neurons occupy the same dorso-ventral area but differ in their anterior to posterior positions within the Abg, with clasper neurons ending more posteriorly than lateral plate neurons (b,c and e,f). See also *Videos 6*,*7*. Hypandrium neurons typically exhibit a unique contralateral arborization pattern within the Abg (d,g). See also *Video 8*. (h) A subset of clasper neurons project to the brain. Unilateral dye-fill of clasper neurons together with extended incubation (10 days) reveals single afferent axon (per hemisphere) that transverses the VNC and terminates in the subesophageal zone (SEZ) of the brain. DiD dye-filled arborizations shown in white. (e-h) DiD dye-filled arborizations shown in white. Boundaries of Abg and brain shown with dotted white line. Afferent projections of dye-filled neurons traced with dotted yellow line. D, dorsal, L, lateral, P, posterior, V, ventral. Scale bar = 25 μm. (i) Arborisations of clasper, lateral plates and hypandrium neurons interdigitate with *dsx/vGlut* dendrites in the adult male Abg. Neurons of all three genital structures were unilaterally dye-filled in males expressing dendritic marker (*UAS-DenMark*) in *dsx/vGlut* neurons. Maximum intensity Z-projection of Abg (i) and 10 μm sub-stack (i') show overt interdigitation (arrowheads) between neurons of all three genital structures and dendrites of *dsx/vGlut* neurons in the Abg. DenMark shown in magenta; DiD shown in green. Scale bar = 25 μm.

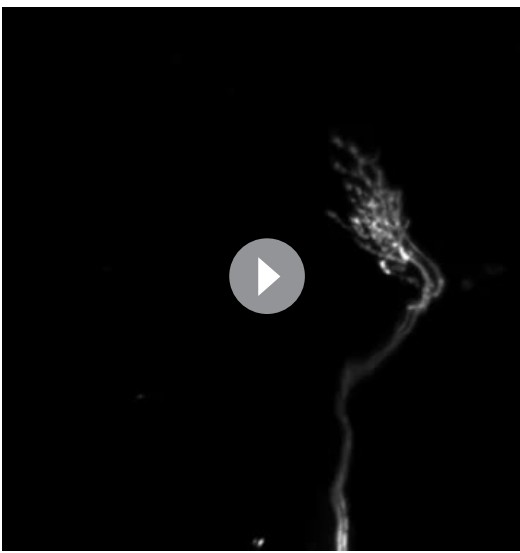

**Video 6.** Clasper neurons of the male genitalia innervate the abdominal ganglion. This movie shows the 3D reconstruction of an adult male abdominal ganglion with innervations of dye-filled neurons from bristles on the clasper of the male genitalia. White: Lipophilic dye (DiD).

## Discussion

### The motor circuit for copulation

This study defines a sexually dimorphic motor circuit in the Abg that mediates the action sequence of copulation in males. We identified three core *dsx*-expressing neuronal types – motor neurons, interneurons and mechanosensory neurons – that control the mechanics of copulation. Excitatory motor neurons promote genital coupling and they are opposed by local inhibitory neurons, which prevent it, while sensory neurons of the genitalia provide sensory feedback to the system to ensure a coordinated sequence of motor events that result in successful copulation (*Figure 8*).

### Control of copulation by interplay of excitation, inhibition and sensory input

The tripartite motor circuit that controls the movement of the genitalia during the initiation and termination of copulation is reminiscent of the spinal microcircuits that coordinate limb movements in mammals (*Miri et al., 2013*). These microcircuits exhibit three features, the first of which involves coordinated firing of motor neurons that trigger the contraction of muscles in the appropriate appendages. *dsx/vGlut* neurons

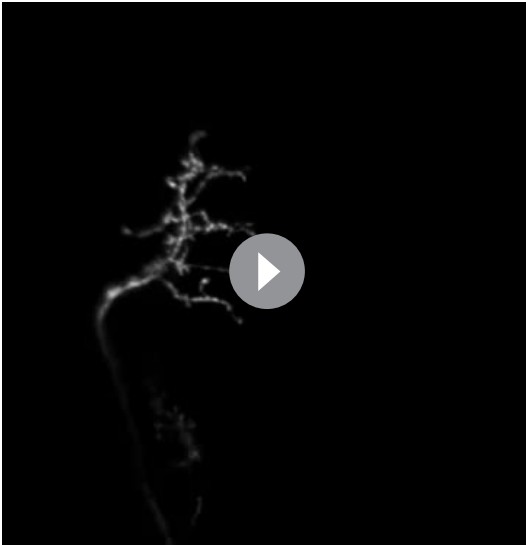

**Video 7.** Lateral plate neurons of the male genitalia innervate the abdominal ganglion. This movie shows the 3D reconstruction of an adult male abdominal ganglion with innervations of dye-filled neurons from bristles on the lateral plate of the male genitalia. White: Lipophilic dye (DiD).

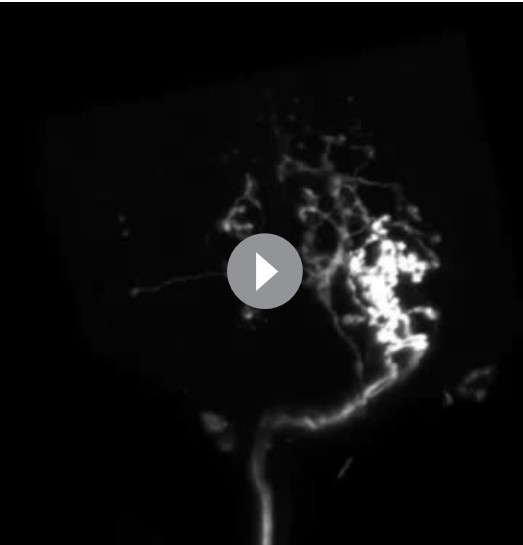

**Video 8.** Hypandrium neurons of the male genitalia innervate the abdominal ganglion. This movie shows the 3D reconstruction of an adult male abdominal ganglion with innervations of dye-filled neurons from bristles on the hypandrium of the male genitalia. White: Lipophilic dye (DiD).

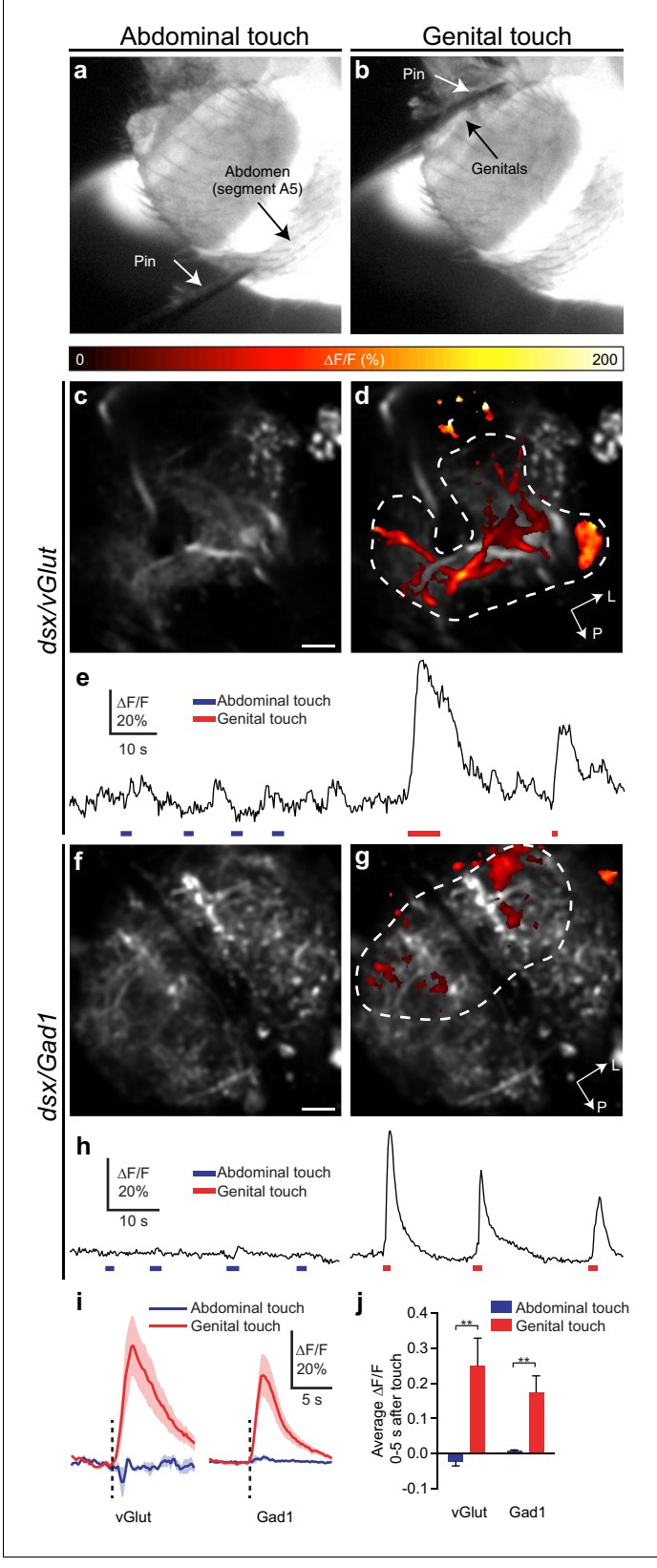

**Figure 7.** Glutamatergic and GABAergic *dsx* neurons of the Abg respond to mechanical stimulation of genitalia. (a,b) Examples of pin touching a male fly's abdomen on (a) segment A5 and (b) genitalia. The fly is illuminated at the VNC by the 910 nm two-photon laser and imaged with an infrared-sensitive camera. (c,d) *dsx/*

*Figure 7 continued on next page*

*Figure 7 continued*

*vGlut>GCaMP6m* neuropil in the abdominal ganglion: Pseudocolored activity maps of responses to (**c**) abdominal or (**d**) genital touch, overlaid on grayscale baseline fluorescence See also *Video 9*. There is no response to abdominal touch (**c**). Dotted outline indicates region of interest for panel E. L, lateral, P, posterior. (**e**) ΔF/F of the outlined region in panel D. Bars under traces represent abdominal (blue) or genital (red) touch. The two traces come from a single movie. (**f**,**g**) as with (**c**,**d**) but for *dsx/Gad1>GCaMP6m* neuropil. See also *Video 10*. (**h**) as with (**e**) but referring to the outlined region in (**g**). (**i**) Average of ΔF/F traces as in (**e**) and (**h**), aligned to touch onset. *dsx/vGlut* and *dsx/Gad1* neurons in the abdominal ganglion respond strongly to genital touch (red) but not abdominal touch (blue). Traces: average ΔF/F (fluorescence normalized to baseline); shading, S.E.M.; vertical line, onset of touch. Only touches < 3 s long are included. n = 7 (vGlut genital), 5 (vGlut abdominal), 6 (Gad1 genital and abdominal). (**j**) Average ΔF/F 0–5 s after onset of touch is significantly larger for genital touch than abdominal touch. ** p<0.01, Mann-Whitney test. Scale bars, 10 μm.

encompass all motor neurons innervating the genital muscles, including both protractor and retractor muscles associated with the phallic and periphallic organs (*Figure 8a*) (*Kamimura, 2010*). Triggering the contraction of these muscles in a sequence-specific manner allows the male to clasp to the female's oviscape prior to protracting the aedeagus and achieving intromission (*Figure 8a,b*). Prematurely stimulating these contractions likely prevents attachment entirely, much as a human hand cannot grasp an object if it is clenched in a fist. Such a scenario would explain why thermoactivating all *dsx/vGlut* neurons blocks copulation, rather than stimulating it, while thermoactivating all *dsx/vGlut* neurons *in copulo* blocks termination of copulation (*Figure 2*). While the *dsx/vGlut* population may also include interneurons, this possibility is unlikely to affect the direct artificial activation and blockade of *dsx/vGlut* motor neurons, which are the final output of the circuit.

The second critical feature of mammalian motor circuits is inhibition of motor neurons by local interneurons, which facilitates the initiation and coordination of limb movement. Here, *dsx/Gad1* neurons likely inhibit *dsx/vGlut* neurons both to terminate copulation and to ensure successful copulation (*Figure 8b*). Copulation termination requires both *dsx/Gad1* synaptic transmission (*Figure 3*) and GABA$_B$ receptor expression in *dsx/vGlut* neurons (*Figure 4*), suggesting that *dsx/Gad1* neurons terminate copulation by metabotropically inhibiting motor neurons. This is supported by the finding that activation of *dsx/Gad1* neurons *in copulo* results in immediate termination of copulation (*Figure 3*). *dsx/Gad1* interneurons may also prevent males from protracting their copulatory organs indiscriminately. These could be inhibited by descending signals to promote copulation or activated by competing drives upon insemination when continuing copulation is no longer necessary (*Figure 8b*) (*Crickmore and Vosshall, 2013*).

Aside from termination, blocking *dsx/Gad1* neurons also reduces successful copulation (*Figure 3*), suggesting that *dsx/Gad1* neurons also modulate the timing of motor neuron activity to achieve the correct sequence of muscle contractions required for copulation, perhaps through reciprocal inhibition (*Figure 8b*). Although we observed a 'stuck' phenotype by knocking-down the GABA$_B$-R2 receptor in *dsx/vGlut* neurons, the role of GABA reception in these neurons is far from resolved (*Figure 4—figure supplement 1*). We would be surprised if GABA$_A$ receptors were not involved since the coordination of fast motor behavior would normally involve fast-responding GABA receptors. These RNAi results may be due to incomplete knockdown and/or homeostatic adaptation (*Lin et al., 2014*). A direct physiological test of how *dsx/Gad1* neurons affect motor neuron activity during copulation awaits new techniques for recording neural activity in the Abg in a copulating male.

The final feature of motor circuits is sensory input, which is required to facilitate fine motor coordination. Sensory neurons of the external genitalia likely provide the initial computations about orientation by extracting salient tactile features of the female that lead to appropriate genital attachment, *e.g.*, during attempted copulation, an objectively purposeful behavior, the male actively acquires sensory inputs ('active sensing') about the female genitalia. These neurons feed forward to *dsx/vGlut* neurons, *dsx/Gad1* interneurons, and to neurons in the brain, suggesting that sensory feedback signals not only local motor circuits in the abdominal ganglion but also higher-level control of male sexual behavior by the brain. Perhaps a function of this input *in copulo* is to aid the establishment of a dynamic balance of excitation and inhibition that mediates the appropriate positioning of the male throughout an entire copulation event.

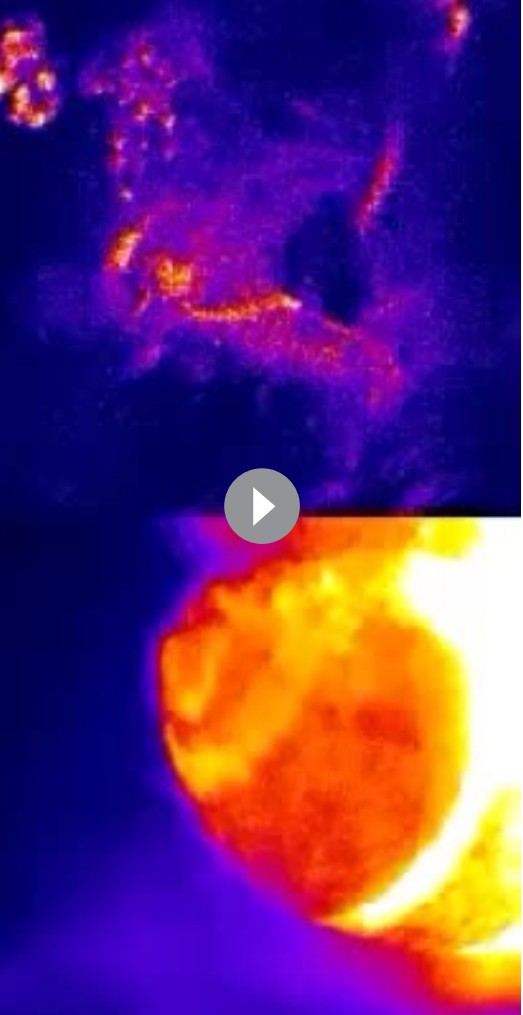

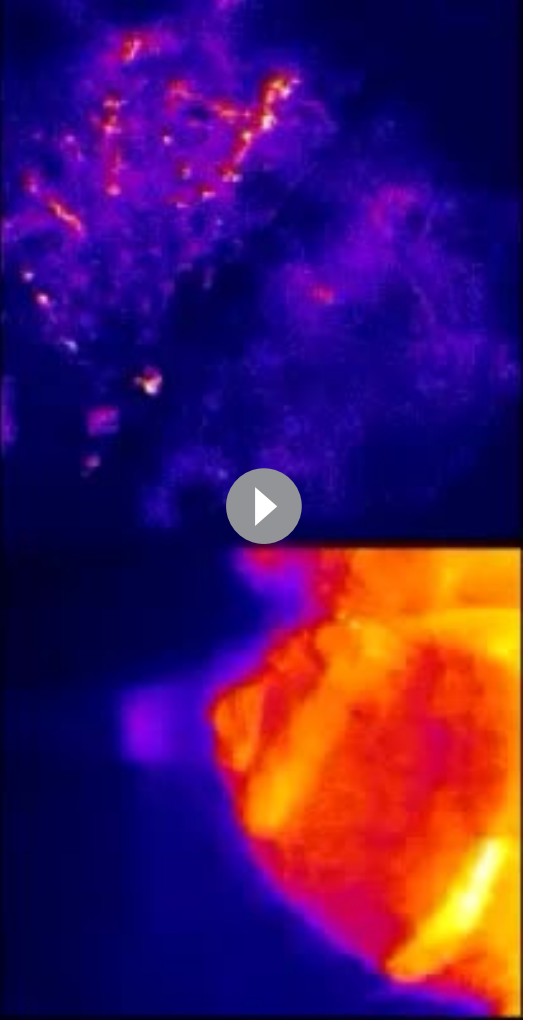

**Video 9.** *dsx/vGlut* Abg neurons respond to mechanical stimulation of genitalia. The upper panel shows GCaMP6m signal in *dsx/vGlut* neuropil in the Abg and the lower panel shows the simultaneous view of the fly's abdomen, illuminated by the 910 nm laser used for two-photon imaging. The minutien pin touches the genitalia at 0:05 and 0:12 depicted with ***. Movies are 5x actual speed and false colored.

**Video 10.** *dsx/Gad1* Abg neurons respond to mechanical stimulation of genitalia. The upper panel shows GCaMP6m signal in *dsx/Gad1* neuropil in the Abg and the lower panel shows the simultaneous view of the fly's abdomen, illuminated by the 910 nm laser used for two-photon imaging. The minutien pin touches the genitalia at 0:04 and 0:08 depicted with ***. Note that the pin approaches the genitalia but does not quite touch until 0:04. Movies are 5x actual speed and false colored.

If *dsx/vGlut* neurons and *dsx/Gad1* neurons oppose each other, then why do subsets of both populations respond to genital touch (*Figure 7*)? One explanation might be that the population of *dsx/vGlut* neurons activated by genital touch might not be inhibited by the population of *dsx/Gad1* neurons that are also activated by such stimulation. Alternatively, sensory excitation of *dsx/vGlut* neurons might outweigh inhibition from *dsx/Gad1* neurons, or some *dsx/vGlut* neurons might indeed be inhibited by genital touch (via *dsx/Gad1* neurons), but inhibition of silent neurons would not give a GCaMP response. Addressing this question awaits the discovery of new driver lines that label distinct subsets of *dsx/vGlut* and *dsx/Gad1* neurons and enable simultaneous imaging, and the development of an image registration methodology that is capable of precise identification of individual neurons in the highly dense Abg between imaging preparations.

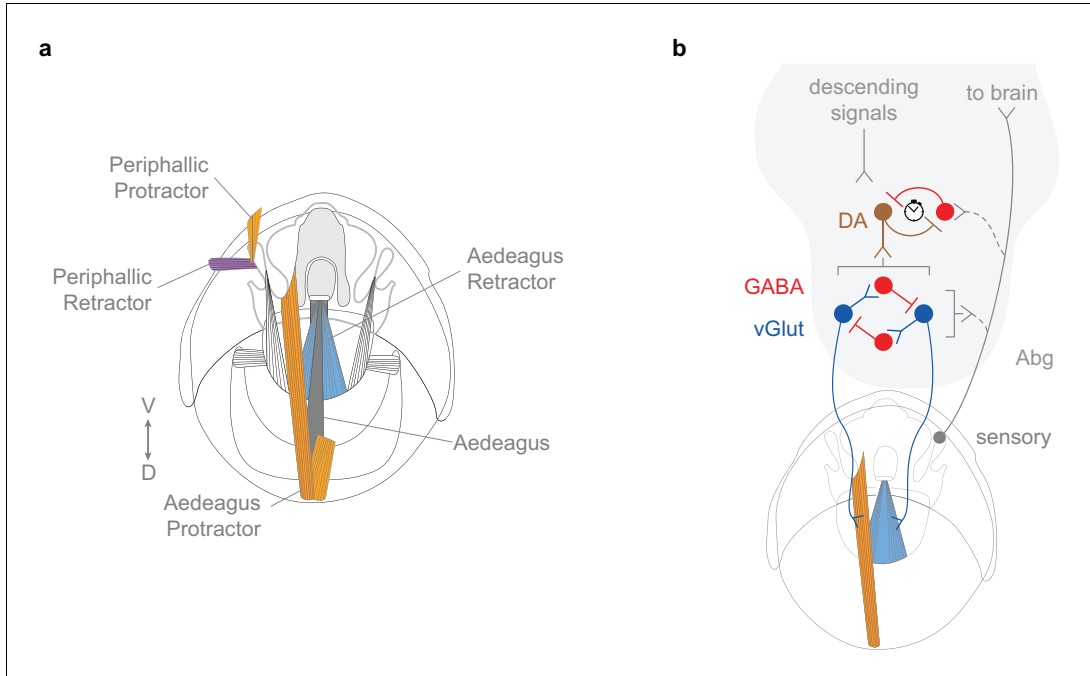

**Figure 8.** Model of circuit organization underlying copulation in males. (**a**) Musculature of male genitalia and terminalia involved in copulation. Protractor muscles shown in orange. Retractor muscles shown in blue. Muscles with no designated colour have unknown functions. D: dorsal, V: ventral. (**b**) *dsx/vGlut* motor neurons local to the Abg (blue) mediate genital coupling by controlling muscles of the phallic and periphallic organs. *dsx/Gad1* Abg neurons (red), depicted as a heterogeneous population of neurons, some of which inhibit glutamatergic neurons that control copulatory muscles (bottom), and other which shorten copulation duration (stopwatch) by reducing copulation motivation by inhibiting dopaminergic (DA) neurons (top). *dsx* sensory neurons of the genitalia (grey) innervate the Abg and brain, and are anatomically and functionally connected to *dsx/vGlut* motor neurons (blue) and *dsx/Gad1* inhibitory centres (red) of the Abg, likely aiding the male in adopting the correct posture to successfully achieve copulation. Brackets depict control over all encompassing neurons.

## Neuronal dissociation of copulatory behaviors

Male copulation is made up of several distinct features: genital coupling, sperm transfer, and copulation duration (which is thought to be a product of declining 'persistence') (*Crickmore and Vosshall, 2013*). Here, we have dissociated genital coupling from all other features; activating *dsx/vGlut* neurons *in copulo* or silencing *dsx/Gad1* neurons produces a 'stuck' phenotype, where males successfully transfer sperm and attempt to terminate copulation at the appropriate time, but cannot uncouple from the female. A recent study has shown that sperm transfer is mediated by corazonin-expressing neurons activating serotonergic neurons that innervate the accessory glands (*Tayler et al., 2012*). Our results show that these neurons are distinct from *dsx/Gad1* and *dsx/vGlut* neurons and that sperm transfer does not require (although it may still be modulated by) *dsx/Gad1* neurons or *dsx/vGlut* neurons.

In addition, we demonstrate a dissociation of genital coupling and copulatory 'persistence' at the neuronal level. A recent study identified ~8 putative *dsx*/GABAergic neurons that control copulation 'persistence' together with dopaminergic neurons in the VNC (DA; *Figure 8b*) (*Crickmore and Vosshall, 2013*). Blocking these eight putative *dsx*/GABAergic neurons does not affect genital uncoupling but rather results in abnormally long copulation durations, after which males detach without struggle (*Crickmore and Vosshall, 2013*). Here, blocking the full complement of 150 *dsx/Gad1* neurons does not affect copulation persistence but rather prevents genital uncoupling, evidenced by the 'stuck' phenotype. This discrepancy can be explained if the full complement of *dsx/Gad1* Abg neurons described here are a heterogeneous population of neurons, some that shorten copulation by reducing copulation motivation and others that inhibit glutamatergic neurons that control

copulatory muscles (*Figure 8b*). Aberrant motor neuron activity patterns caused by blocking the latter might change sensory or internal feedback to the dopaminergic neurons controlling copulation persistence, explaining why blocking all *dsx/Gad1* neurons causes a 'stuck' phenotype rather than increasing copulation persistence. Such 'specialist' inhibitory interneurons are present in many mammalian spinal circuits that drive complex movements (*Graziano, 2006*). Our results reveal that distinct populations of *dsx*/GABAergic neurons likely control different features of copulation (genital coupling and persistence).

### Initiation and modulation of copulation

If the Abg can act autonomously to control copulatory behaviors (*Billeter et al., 2006*; *Pan et al., 2011*; *Tayler et al., 2012*; *Crickmore and Vosshall, 2013*; *Inagaki et al., 2014*), then what role does the brain play? In mammals, brain circuits are thought to compute adaptive neural functions that underlie action-selection (or 'decision-making'), while spinal circuits communicate bidirectionally with the brain, such that descending pathways activate motor programs and ascending pathways report on their execution (*Arber, 2012*). This organizing principle also exists in the fly: male-specific 'decision-making' neurons in the brain (e.g., P1; pC2l; dopaminergic neurons) integrate multimodal sensory cues to initiate and modulate courtship and copulatory behaviors (*Kimura et al., 2008*; *Yu et al., 2010*; *Kohatsu et al., 2011*; *von Philipsborn et al., 2011*; *Pan et al., 2012*; *Inagaki et al., 2014*; *Kohatsu and Yamamoto, 2015*; *Zhang et al., 2016*), while descending 'command' neurons (e.g., pIP10) relay information from the 'decision-making' neurons, to activate 'pattern generator'-like motor circuits in the VNC (*Clyne and Miesenbock, 2008*; *von Philipsborn et al., 2011*; *Inagaki et al., 2014*).

We similarly predict that higher-order neurons in the brain control the initiation and modulation of the copulation motor circuit described here. The culmination of the courtship ritual would activate 'decision-making' neurons that signal, via descending 'command' neurons, to *dsx* glutamatergic and GABAergic neurons in the Abg, to initiate genital coupling. Meanwhile, descending and ascending signals from these neurons (including sensory feedback from the genitalia) would modulate the activity of the motor circuit in response to the male's internal state and the environment. Indeed, males mate for longer and increase their sperm load when in the presence of rival males (*Bretman et al., 2009*; *Garbaczewska et al., 2013*). This behavioral responsiveness most likely relies on physiological modulation of 'decision-making' neurons, much as increased courtship behavior in socially isolated male flies correlates with increased excitability of 'decision-making' neurons for courtship (*Inagaki et al., 2014*). In addition, while the *dsx/vGlut* and *dsx/Gad1* populations are sexually dimorphic, it is interesting to speculate that similar organizational principles may govern female reproductive behavior, with the analogous female *dsx/vGlut* and *dsx/Gad1* neurons controlling behaviors like ovipositor extrusion and retraction. Our findings provide insight into the circuit logic underlying genital coupling, and serve as an entry point to a circuit-level dissection of copulatory behavior and its modification by social experience.

## Materials and methods

### Targeted insertion of GAL4-DBD into the *dsx* locus

GAL4-DBD was targeted to the *dsx* locus by ends-in homologous recombination as previously described (*Rideout et al., 2010*). In a single cloning step, the GAL4-DBD coding sequence was excised from pCaST-elavGAL4-DBD (gift from B. White) with BglII and BamHI restriction enzymes, and ligated into the original *dsx*$^{GAL4}$ construct after digestion with BamHI. PCR was used to confirm the predicted recombination event in multiple lines. Two lines were selected for further analysis using multiple *UAS* reporter transgenes and both exhibited equitable patterns of expression (data not shown), of which a single *dsx*$^{GAL4-DBD}$ line was then chosen for this study. *dsx*$^{GAL4-DBD}$ not a *dsx* null mutant; *dsx*$^{GAL4-DBD}$ homozygotes are fertile and show normal morphology (data not shown).

### Fly strains

All flies were raised at 25°C on standard medium in a 12 hr light/12 hr dark cycle at 50% relative humidity. Fly strains used in this study include wild-type *Canton-S*; *dsx*$^{GAL4-DBD}$ (generated in this study); *dsx*$^{GAL4}$ (*Rideout et al., 2010*); *elav*$^{VP16-AD}$ (*Luan et al., 2006*); *vGlut*$^{OK371-dVP16-AD}$ denoted

as $vGlut^{dVP16-AD}$ (*Gao et al., 2008*); $Gad1^{p65-AD}$ (*Diao et al., 2015*); UAS-TNT$_G$ (*Sweeney et al., 1995*); UAS-dTrpA1 (*Hamada et al., 2008*); UAS-pStingerII denoted as UAS-nGFP (*Barolo et al., 2000*); UAS-2XEGFP (*Halfon et al., 2002*); UAS-DenMark (*Nicolaï et al., 2010*); UAS-nSyb::GFP (*Estes et al., 2000*); UAS-GCaMP6m (*Chen et al., 2013*); UAS>stop>myrGFP (*Yu et al., 2010*); UAS>stop>TNT and UAS>stop>dTrpA1$^{myc}$ (*von Philipsborn et al., 2011*); Otd-nls:FLPo (*Asahina et al., 2014*); UAS-GABA$_B$-R1-RNAi (VDRC: 101440); UAS-GABA$_B$-R2-RNAi (VDRC: 1784); UAS-GABA$_B$-R3 (Bloomington: 26729); UAS-Lcch3-RNAi (VDRC: 37408) and UAS-Rdl-RNAi (*Liu et al., 2007*). All lines and transgenes were in a $w^+$ background for behavioral studies.

## Behavior

Crosses with Split-GAL4 and *UAS-effectors* were raised at 21°C in a 12:12 hr light:dark cycle and at 50% relative humidity. Crosses containing *Otd-nls:FLPo*, Split-GAL4 and *UAS>stop>effectors* transgenes or RNAi transgenes were raised at 25°C in a 12:12 hr light:dark cycle and at 50% relative humidity.

### Neuronal silencing and RNAi knockdown

UAS-TNT, UAS>stop>TNT and RNAi expressing flies were aged at 25°C in a 12:12 hr light:dark cycle. Individual virgin males were collected and aged for 3–5 days post-eclosion while virgin females were aged for 5–7 days post-eclosion at 25°C. Courtship assays were carried out at 25°C where individual females were introduced into a round courtship chamber (19 mm diameter × 4 mm height) with an individual naïve control or experimental male.

### Neuronal thermoactivation

UAS-dTrpA1 expressing flies were aged at 21°C in a 12:12 hr light:dark cycle. Individual virgin males were collected and aged for 5–7 days post-eclosion while virgin females were aged for 7–9 days post-eclosion at 21°C. UAS>stop>dTrpA1$^{myc}$ expressing flies were aged at 23°C in a 12:12 hr light: dark cycle. Individual virgin males were collected and aged for 4–6 days post-eclosion while virgin females were aged for 6–8 days post-eclosion at 23°C.

Courtship assays were carried out at the permissive temperature of 23°C, or the restrictive temperature of 31°C. Single females were introduced into a round courtship chamber (20 mm diameter × 4 mm height) with a single naïve control or experimental male. Males and females were kept separate for the first 10 min so that the chambers to reach the precise temperature of the underlying heating plate; after which the separator was removed and the flies were introduced.

To thermoactivate mating pairs during copulation, individual pairs of males and females were placed in small chambers (10 mm diameter x 4 mm height) at the permissive temperature of 23°C and monitored for copulation. Upon initiation of copulation, mating pairs were timed for 5 min and then shifted to a heating block set at restrictive temperature of 31°C. The subsequent time it took to terminate copulation (in sec) was recorded.

### Behavioral parameters

Parameters of male courtship behavior are as previously described (*Rideout et al., 2010*). *Courtship index* (*CI*) is measured as the proportion of time in 10 min that the male spent exhibiting courtship behaviors towards the 'target' female. Courtship behaviors are defined here as: following, orientation, tapping, wing extension, and abdominal curling/attempted copulation. *Copulation duration* is a measure of the observed time (in sec) elapsed between the beginning of a copulation event and its termination. *% Mating in 1 hr* is proportion of male flies (as a percentage) that successfully copulated within in a 1 hr period. *% Fertile matings* is the proportion of copulation events that produce viable progeny. Individual females from successfully mated pairs were placed individually in food vials. All vials were scored for presence of progeny after 10 days. *% Copulations terminated* is the cumulative proportion of copulating pairs that terminate copulation over time. *% Fertility* is the proportion of male flies that produce viable progeny. Males tested for fertility were collected at eclosion (kept in groups ≤ 10) and aged for five days. They were subsequently placed individually in food vials containing three wild-type virgin females of the same age. All vials were scored for presence of progeny after 10 days. Vials containing a dead male or female and no progeny were discounted.

Parameters of female behaviors are as per previously described (*Rideout et al., 2010*). *% Fertility* is the proportion of females that produce viable progeny. Females tested for fertility were collected at eclosion (kept in individual vials) and aged for five days. They were then introduced individually into food vials containing three wild-type virgin males of the same age. All vials were scored for presence of larval progeny after 10 days. Vials containing a dead female or male and no progeny were discounted. *Line crossings* is a measure of the number of times a copulating pair crossed a demarcated line in the courtship chamber (per min) during the time spent copulating. *Copulation duration* is a measure of the observed time (in sec) elapsed between the beginning of mating and its termination. *% Re-mating* is the proportion of females, as a percentage, that were successfully re-mated by the same wild-type male over the same 1 hr period in which females mated for the first time. *'Unreceptive females'* characterizes a syndrome in which females are incapable of sampling the male's display and providing any acceptance response. Although this does not prevent them from copulating, copulating these females lack cooperation by continuously moving and displaying rejection behaviors. Successful copulation is therefore a testament to the male's persistence, rather than the female's receptivity.

## Immunohistochemistry

Flies were reared at 25°C and aged for 4–6 days prior to dissection and staining as previously described (*Rideout et al., 2010*). Samples were dissected in PBS and fixed in 4% (w/v) paraformaldehyde (in PBS) for 20 min at room temperature (RT). Primary antibody incubation was carried out for 24–48 hr at 4°C. Secondary antibody incubation was carried out for 24 hr at 4°C.

Primary antibodies used included: rabbit anti-GFP (1:1000, Invitrogen Molecular Probes, Carlsbad, CA), chicken anti-GFP (1:1000, Abcam, UK), rabbit anti-DsRed (1:1000, Clontech); mouse mAb nC82 (1:10, DSHB, Univ. of Iowa, IA), rabbit anti-βGal (1:1000; Cappel, ICN), rabbit anti-Fru$^M$ at 1:3000 (*Billeter et al., 2006*); rabbit anti-dvGlut at 1:500 (*Mahr and Aberle, 2006*) and rabbit anti-GABA (1:2000, Sigma-Aldrich). Secondary antibodies used included: anti-chicken Alexa Fluor488, anti-rabbit Alexa Fluor488, anti-rabbit Alexa Fluor546, anti-mouse Alexa Fluor546, anti-rat Alexa Fluor546, anti-mouse Alexa Fluor633, anti-rat Alexa Fluor633, anti-rat Cy5 (1:300, Invitrogen Molecular Probes, Carlsbad, CA), HRP-Cy3 conjugate (1:100, Sigma-Aldrich) and Phalloidin-TRITC and -Alexa Fluor633 conjugates (1:100, Sigma-Aldrich).

Samples were mounted with Vectashield (Vector Labs) and imaged with an Olympus FluoView FV1000 confocal microscope x10 (air), x20 (air), x40 (oil immersion), and x63 (oil immersion) objectives. For multi-track (multiple fluorophore labels) imaging, each wavelength was sequentially scanned for each optical section through the sample to excite each fluorophore individually and avoid bleed-through. Stacks of optical sections were generated at 1 μm intervals. Images were processed in Imaris (Bitplane Scientific, AG, Zürich) and peripheral debris was removed in Adobe Photoshop 7.0. (Adobe Systems Inc., San Jose, CA).

For cell counts, stacks of optical sections obtained by confocal microscopy were transformed into maximum Z-stack projections in Imaris. The fluorophore labeling cells of interest was used as the 'source channel' for automatic detection of spherically labeled nuclei with a 3 μm minimum in diameter using the Imaris 'Spots' detection module. To ensure that all obvious cells were marked, the threshold was manually shifted to the point at which it automatically detects the maximum number of spherical nuclei, without any observable ectopic detection. Subsequent use of the 'orthoslicer' tool allowed for examination of each optical slice. Un-marked nuclei that fell short of automatic-detection were manually marked. A final count of the number of marked nuclei was subsequently calculated.

For brain image registration, confocal images of male *dsx/vGlut>nSyb* and *dsx/vGlut>DenMark* VNCs were registered onto a *D. melanogaster* intersex template VNC that was generated by the Jefferis lab (http://zenodo.org/record/10591#), using a Fiji graphical user interface (GUI), and the previously described (*Jefferis et al., 2007*; *Cachero et al., 2010*; *Ostrovsky et al., 2013*). 2 VNCs per genotype were used for the analysis.

## Retrograde dye-fills

The protocol to label the complex axonal branching of mechanosensory neurons of the male genitalia with a lipophilic fluorescent dye is adapted from established protocol (*Kays et al., 2014*). Adult

(2–3 day old) male flies were perpendicularly glued onto insect pin heads, decapitated (or not) and fixed in 3.7% paraformaldehyde in in 0.2 M carbonate-bicarbonate buffer at pH 9.5 overnight at 4°C. Flies were subsequently washed with ddH$_2$O and gently dried with a Kimwipe tissue. Mechanosensory neurons of the lateral plates, claspers and hypandrium were each or all unilaterally dyed by subcuticular injection of DiD dye (80 µg/µL in 100% ethanol; Life Technologies, cat. no. D7757) using a micropipette and micromanipulator (settings for a Sutter P-97 Flaming/Brown micropipette puller using standard borosilicate glass of o.d./i.d. 1.00 mm/0.78 mm with filament are as follows: heat = 515, pull = 30, velocity = 30, time = 165). Pins with attached dye-filled flies were wedged into clay such that the thorax of the flies was submerged in 0.2 M carbonate-bicarbonate buffer while their abdomen and genitals were left above the surface of the buffer. Samples were incubated in the dark for six days to realize innervations in the Abg and 10 days to realize innervations in the brain. CNSs and VNCs were dissected in PBS and imaged with an Olympus FluoView FV1000 confocal microscope within 10 min of mounting in Vectashield (Vectorlabs).

## Two-photon calcium imaging

1–3 day old male flies were waxed to tin foil in a perfusion chamber such that the ventral thorax faced the objective through a small hole in the foil. The legs, cuticle and trachea covering the abdominal ganglion were removed and the preparation was superfused with solution (in mM: TES 5, NaCl 103, KCl 3, CaCl$_2$ 1.5, MgCl$_2$ 4, NaHCO$_3$ 26, NaH$_2$PO$_4$ 1, trehalose 8, and glucose 10, pH 7.3) bubbled with carbogen (95% O$_2$, 5% CO$_2$). The genitalia or abdomen were mechanically stimulated using a 0.1 mm stainless steel insect pin (Fine Science Tools 26002–10) attached by a thin rod to a manual micromanipulator (Märzhäuser Wetzlar MM-33). Stimulation of the genitalia or abdomen was recorded at 4.07 Hz with a Stingray F-033B camera (Allied Vision Technologies), using the illumination of the fly from the 910 nm laser during two-photon imaging.

Two-photon imaging and data analysis was adapted from established protocol (*Lin et al., 2014*). The abdominal ganglion was imaged at 4.34 Hz with a pixel dwell time of 3.2 µs. The baseline fluorescence (F0) for calculating ΔF/F was defined as the average signal 1–5 s before stimulus onset. For activity maps, we excluded pixels where ΔF (difference between F0 and mean fluorescence in the first 5 s after stimulus onset) was less than twice the standard deviation of fluorescence during the F0 period. Regions of interest (ROIs) were manually drawn around responsive areas and ΔF/F traces were aligned to the start of genital or abdominal touch. To prevent slow-decaying GCaMP6m signals from the previous stimulus from interfering with this baseline calculation, genital stimuli were only used if they occurred more than 10 s after the start of the previous stimulus.

## Statistics

Behavioral means were compared using Kruskal-Wallis ANOVA test and Dunn's post hoc statistical test where indicated. For Fisher's exact test, two-tail *p* values were compared with controls. Statistical analyses were performed with the GraphPad Prism software (version 6.0, GraphPad Software Inc.).

## Acknowledgements

We thank Anthony Dornan, Scott Waddell, and members of the Goodwin lab for helpful discussions and critical reading of the manuscript. We thank Farida Emran and Ibrahim Kays for technical assistance with dye-fills. We thank Janice Ting for kindly generating schematics of the male genitalia. We are grateful to David Anderson, Barry Dickson, and Chi-Hon Lee, for sharing fly lines with us. We also obtained fly lines from Bloomington Stock Center. This work was supported by the Intramural Research Program of the National Institute of Mental Health to BHW (ZIAMH002800), the EP Abraham Cephalosporin Trust Fund to HJP and SFG, the Brain@McGill Collaboration Fund to HJP, BEC and SFG, and a Wellcome Trust Investigator Award to SFG (WT106189MA).

## Additional information

### Funding

| Funder | Grant reference number | Author |
| --- | --- | --- |
| EP Abraham Cephalosporin Trust Fund | | Hania J Pavlou<br>Stephen F Goodwin |
| Brain@McGill Collaboration Fund | | Hania J Pavlou<br>Brian E Chen<br>Stephen F Goodwin |
| National Institute of Mental Health | ZIAMH002800 | Benjamin H White |
| Wellcome Trust | WT106189MA | Stephen F Goodwin |

The funders had no role in study design, data collection and interpretation, or the decision to submit the work for publication.

### Author contributions

HJP, MCN, Conception and design, Acquisition of data, Analysis and interpretation of data, Drafting or revising the article, Contributed unpublished essential data or reagents; ACL, Acquisition of data, Analysis and interpretation of data, Drafting or revising the article; TN, Acquisition of data, Analysis and interpretation of data; FD, BHW, Provided key Split-GAL4 neurotransmitter hemi-drivers, Contributed unpublished essential data or reagents; BEC, Designed the experiments, Conception and design; SFG, Conception and design, Analysis and interpretation of data, Drafting or revising the article

### Author ORCIDs

Stephen F Goodwin, http://orcid.org/0000-0002-0552-4140

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
