## [Decision Letter]

Thank you for submitting your article "Neural circuitry coordinating male copulation" for consideration by *eLife*. Your article has been favorably evaluated by K VijayRaghavan (Senior Editor) and three reviewers, one of whom, Mani Ramaswami (Reviewer #1), is a member of our Board of Reviewing Editors.

The reviewers have discussed the reviews with one another and the Reviewing Editor has drafted this decision to help you prepare a revised submission.

Summary:

In this paper, the authors use carefully designed and beautiful split-Gal4 systems to identify sexually dimorphic neurons in the abdominal ganglion that are required for copulation behaviour. They do a very nice job of identifying the three main sets of sexually dimorphic *dsx*+ neurons in the male that are responsible for the execution and termination of copulation. This is a relatively understudied aspect of the entire mating process, so the information provided here breaks new and interesting ground. Glutamatergic motor neurons that facilitate genital attachment, GABA inhibitory interneurons that regulate the motor neurons, and genital sensory neurons, all *dsx*+, were identified and experimentally tested for their putative roles in the circuit. In the end, the work focuses on the neurons in the abdominal ganglia of the ventral nerve cord, where, not surprisingly, much of the action seems to take place. Activation and silencing experiments were conducted for the motor neurons and GABA inhibitory neurons, using powerful splitGal4 genetic tools. Given the absence of such tools for analogous experiments with the sensory neurons, the authors demonstrate GCaMP activation of the motor and GABA neurons in response to genital stimulation, to justify a clear and scholarly model for the circuit logic involved in regulation of copulatory behaviour. The Discussion was a pleasure to read, and nicely put this work into the broader perspective.

There are several firsts associated with this manuscript. It is the first description of a microcircuit proposed to control the male organ during sex, first description of sensory bristles acting to relay sensory information to neurons involved in copulation, first demonstration that the mechanics of copulation can be separated from the control of ejaculation in the fly. It is further noteworthy that the interneurons described here appear to play a different functional role from those described by Crickmore in another outstanding paper. Crickmore's work demonstrated interneurons that control dopaminergic circuitry and regulate the timing of copulation. The Crickmore interneurons could exert an influence by modulating the circuitry described here (by mechanisms unknown). This paper provides a bottom line for any other study that seeks to understand how copulatory behaviour is modulated. Pavlou and colleagues’ discovery and presentation provide a valuable caution against formulaic expectations based on the idea that one or a couple of neurons controls a behaviour. This study retains the complexity of the nervous system while providing a detailed picture of a reflex arc that controls mating behaviour. The ability to separate copulatory behaviour from ejaculation is also of interest because it suggests a mechanism for separating hedonic features of copulation from reproductive function but the authors avoid this sort of speculation. Overall, Pavlou and co-workers deserve credit for an interesting study and a major contribution to the literature.

The authors should revise the text to address the following issues clearly:

1) The use of *UAS-TNT* instead of *UAS-Shi(ts)* in several experiments. The precision of the behavioral observations argues against any complexities caused by secondary developmental defects but it is somewhat surprising and the reasoning for the choice of inhibitory transgene is not explicitly discussed.

2) The RNAi experiments with GABA receptors are not conclusive with regards GABA receptor subunits whose attempted knockdown have no effects on behavior. These data (particularly Figure 4—figure supplement 1) are probably the least conclusive of all presented in the paper. It would be strange if GABA-A receptors were not involved as for coordination of fast motor behaviour, one would normally expect involvement fast GABA receptors to be critical. I think clarification and acknowledgement of the limitations of this analysis is required.

3) Although the behavioral consequences of motor neuron activation were well characterized in the paper, and the presumptive muscles were identified that these motor neurons project to, are there any anatomically visible consequences to motor neuron activation/muscle contraction?

---

## [Author Response]

*The authors should revise the text to address the following issues clearly:*

*1) The use of UAS-TNT instead of UAS-Shi(ts) in several experiments. The precision of the behavioral observations argues against any complexities caused by secondary developmental defects but it is somewhat surprising and the reasoning for the choice of inhibitory transgene is not explicitly discussed.*

We thank the reviewers for raising this concern. We did generate the necessary stock *dsx*^GAL4-DBD^, *UAS-Shi(ts)* for neuronal silencing experiments, however problems with the general health of this stock precluded its use. The alternative choice of *UAS-TNT* did not impinge on the health of the fly, and presented the best workable option for these experiments. It is important to note that we took the precaution of raising flies that express *UAS-TNT* at a lower temperature; using this strategy, we and others have been able to limit potential secondary developmental defects (Examples include, but are not limited to: Rideout et al., 2010; Rezával et al., 2012, 2014, 2016; and Crickmore and Vosshall, 2013). We observed no gross anatomical abnormalities in the genitalia or reproductive system, including their neuronal innervation (data not shown).

*2) The RNAi experiments with GABA receptors are not conclusive with regards GABA receptor subunits whose attempted knockdown have no effects on behavior. These data (particularly Figure 4—figure supplement 1) are probably the least conclusive of all presented in the paper. It would be strange if GABA-A receptors were not involved as for coordination of fast motor behaviour, one would normally expect involvement fast GABA receptors to be critical. I think clarification and acknowledgement of the limitations of this analysis is required.*

We agree with the reviewers about the limitations of the GABA receptor knockdown experiments, and the unsatisfying nature of the conclusions. This point has now been addressed in the Discussion.

“[…] Although we observed a ‘stuck’ phenotype by knocking-down the GABAB-R2 receptor in *dsx/vGlut* neurons, the role of GABA reception in these neurons is far from resolved. We would be surprised if GABAA receptors were not involved since the coordination of fast motor behavior would normally involve fast-responding GABA receptors. These RNAi results may be due to incomplete knockdown and/or homeostatic adaptation (Lin et al., 2014).”

*3) Although the behavioral consequences of motor neuron activation were well characterized in the paper, and the presumptive muscles were identified that these motor neurons project to, are there any anatomically visible consequences to motor neuron activation/muscle contraction?*

This is an interesting point and one we are investigating. We did see that in a solitary male, thermal activation of *dsx/vGlut* neurons resulted in no observable protraction of the aedeagus, possibly due to the simultaneous activation of antagonistic muscle groups. This is supported by our findings that thermal activation of *dsx/vGlut* neurons prohibits males from achieving genital attachment with a female (Figure 2). At present, we are developing novel tools to investigate the precise neuromuscular architecture and physiology of the aedeagus.